**EMBO** *reports*

# Acute exogenous acyl-GIP treatment enhances lipid handling and fatty acid oxidation by involving brown fat

Sulayman A Lyons[1,6], Micah B S Lea [1,6], Mihir Parikh[1], Zhengzhang Guo [1], Samrin Kagdi[1], Abigail R Bisnauth[1], Jonathan R Pitino[1], Sabrina Ziai[1], Negar Mir[1], Aidan D Tyrrell[1], Yan Fu [1], Chuck T Chen [1], Adam H Metherel[1], Richard P Bazinet[1], Bin Yang[2,3], Patrick J Knerr[2,4], Jonathan D Douros[2,4], Jonathan E Campbell [5] & Jacqueline L Beaudry [1✉]

## Abstract

**The contribution of glucose-dependent insulinotropic polypeptide receptor (GIPR) signalling in brown adipose tissue (BAT) remains underexplored. We studied the acute effects of exogenous acyl-GIP (1 nmol/kg) administration on whole-body lipid handling and fatty acid oxidation, using lipid tolerance tests (LTT) and indirect calorimetry, respectively. We demonstrate that in obese male mice, acute acyl-GIP administration improves lipid tolerance; however, pharmacological inhibition of GIPR, or genetic removal of GIPR globally or with the *Myf5*-Cre driver, completely abolishes GIP-mediated improvements in lipid tolerance, implicating GIPR in BAT. GIP-mediated improvements in lipid tolerance are associated with an increase in BAT lipid uptake, linked to increases in BAT lipo-protein lipase activity. Our data also reveal that BAT GIPR signalling is necessary for GIP-mediated increases in whole-body fatty acid oxidation, as *Myf5*-Cre: *Gipr* mice do not shift substrate oxidation upon GIP administration. Our findings suggest that BAT should be more closely considered in studies examining GIP's effects on whole-body metabolism in rodent models.**

**Keywords** Glucose-dependent Insulinotropic Polypeptide; Lipid Metabolism; Brown Adipose Tissue
**Subject Category** Metabolism

## Introduction

The inclusion of glucose-dependent insulinotropic polypeptide receptor (GIPR) agonist-based medications has been very successful for the treatment of type 2 diabetes (T2D) and obesity (Aronne et al, 2024; Gallwitz, 2022; Kusminski et al, 2024; Lyons and

Beaudry, 2023; Müller et al, 2025). GIP has mainly been associated with incretin action, where endogenous release of GIP from intestinal $K^+$ cells binds to GIPRs found in the pancreas to stimulate insulin release following meal ingestion (Oteng et al, 2024; Pederson and McIntosh, 2004). GIPRs are also found in a variety of extra-pancreatic tissues, including the central nervous system and adipose tissue (AT) (Hammoud and Drucker, 2023). Central GABAergic GIPR+ neurons mediate the food intake-reducing effects of both GIPR monoagonists and dual incretin receptor agonists in rodents, leading to weight loss (Zhang et al, 2021; Liskiewicz et al, 2023; Wean et al, 2024). Similarly, reductions in food intake have been linked to increased plasma GIP released from the small intestine, demonstrating the important role of the gut-brain GIP-axis on appetite (Lewis et al, 2024).

GIPR signaling in AT has become a likely candidate for GIPR-based medication improvements in metabolic health (Kagdi et al, 2024; Samms et al, 2020). GIP administration to human white adipocytes increases lipolysis in the fasted state and free fatty acid and glucose uptake in the fed state (Killion et al, 2020; Regmi et al, 2024). Similarly, GIP promotes white adipose tissue (WAT) glucose and fatty acid uptake and improves insulin sensitivity in obese insulin-resistant mice (Killion et al, 2020; Samms et al, 2021). Given that endogenous GIP has been proposed to influence fat deposition into WAT (Møller et al, 2016; Musso et al, 2009; Yu et al, 2025), it is possible that exogenous GIP-mediated fat deposition further increases WAT fatty acid uptake and relieves other peripheral organs from ectopic lipid accumulation. However, the link between these GIP-mediated improvements in whole-body lipid handling and metabolism with AT function remains unclear.

While WAT primarily functions to store and release energy (Trayhurn and Beattie, 2001), brown adipose tissue (BAT) mainly functions to uptake and oxidize metabolic substrates, predominantly fats, to generate heat (Bartelt et al, 2011; Cannon and Nedergaard, 2004; Carpentier et al, 2018; Shinde et al, 2021). Interestingly, GIPR has been found to be expressed in BAT, though to a lower extent than WAT (Beaudry et al, 2019; Hammoud and

[1]Department of Nutritional Sciences, Faculty of Medicine, University of Toronto, Toronto, ON, Canada. [2]Novo Nordisk Research Center, Indianapolis, IN, USA. [3]Dexatide LLC, Plainfield, IN, USA. [4]Indiana Biosciences Research Institute, Indianapolis, IN, USA. [5]Duke Molecular Physiology Institute, Duke University, Durham, NC, USA. [6]These authors contributed equally: Sulayman A Lyons, Micah B S Lea. ✉E-mail: jacqueline.beaudry@utoronto.ca

Drucker, 2023). Mice that lack GIPR in BAT have impaired lipid handling following an oral lipid challenge (Beaudry et al, 2019), and chronic GIP administration has been associated with increased gene expression for lipid, glucose, and branched-chain amino acid catabolism, specifically in BAT (Samms et al, 2021, 2022). In addition, acute GIP treatment to individuals with type 1 diabetes and no onboard insulin showed an increase in supraclavicular BAT activity associated with changes in free fatty acid levels and lower respiratory exchange ratios (RER) indicative of more fatty acid oxidation (Heimburger et al, 2022). How GIP is specifically driving these changes in BAT has yet to be characterized.

Altogether, there is a lack of research elucidating the mechanisms behind exogenous GIP-mediated improvements in whole-body lipid handling and oxidation in vivo. Therefore, the primary objective of this study was to determine how exogenous GIP administration impacts mechanisms of whole-body lipid tolerance and oxidation in obese male mice. With the use of an oral lipid challenge, we discovered that a single dose of acyl-GIP blunts hypertriglyceridemia through mechanisms of plasma lipid clearance into BAT via increases in lipoprotein lipase (LPL) activity. We then found that improvements in oral lipid challenges were abolished in BAT-specific GIPR knockout mice. Furthermore, we discovered that GIPR signaling in BAT was associated with increased whole-body fatty acid oxidation. This study is the first to demonstrate that effects in GIP-mediated lipid handling and oxidation are primarily driven through acute GIPR signaling in BAT, independent of body weight loss and food intake.

## Results and discussion

A lipid tolerance test (LTT) is an effective way of assessing an organism's ability to deal with an acute bout of hypertriglyceridemia, revealing both intestinal and whole-body lipid metabolism (Ochiai, 2020). Compared to phosphate-buffered saline (PBS) treated animals, GIP administration has been associated with improvements in LTT, where triglyceride (TG) excursions are lowered, as well as improving glucose tolerance and insulin sensitivity (Hinke et al, 2002; Mroz et al, 2019; Regmi et al, 2024; Samms et al, 2021). In our study, we placed 8-week-old male C57BL/6J mice on a 60% HFD for at least 12 weeks to induce weight gain and insulin resistance (Fig. EV1A–C). We determined that 1 nmol/kg acyl-GIP was the lowest dose necessary for improving lipid tolerance in obese mice ($P < 0.01$; Fig. EV1D) while also reducing blood glucose during an intraperitoneal glucose tolerance test ($P < 0.0001$) in both obese and lean mice (Fig. EV1E–G). We found that acyl-GIP lowered lipid excursions ~38% compared to PBS-treated mice during a LTT (Fig. 1A,B). We also determined that GIPR agonism was necessary for lowering lipid excursions during an LTT. Pharmacological inhibition of GIPR (1500 nmol/kg GIPR antagonist) eliminated GIP-mediated improvements in lipid tolerance (Fig. 1C) and glucose tolerance (Fig. EV1H). Similarly, germline removal of GIPR ($Gipr^{-/-}$ mice) eliminated GIP-mediated improvements in lipid tolerance ($P = 0.0307$, $P = 0.5971$; Fig. 1D,E, respectively). Interestingly, despite showing improvements in GIP-mediated glucose tolerance in obese mice (Fig. EV1F,G), the effects of acyl-GIP on LTT performance were absent in lean regular chow-fed mice ($P > 0.05$; Fig. EV1I,J). There were no differences in AT *Gipr* mRNA

expression between RCD and HFD-fed mice ($P < 0.05$; BAT, eWAT, rWAT), aside from inguinal WAT (iWAT) where *Gipr* expression significantly increased with HFD feeding ($P = 0.0165$; Fig. EV1K). Altogether, our data suggest that the responsiveness of an organism to a specific dose of acyl-GIP during a LTT could be dependent on both GIPR agonism and the metabolic status of an animal (i.e., lean vs obese).

The lipid-lowering mechanisms of acyl-GIP during a LTT could be the result of either a lack of lipids entering the systemic circulation, and/or an increase in lipid uptake into peripheral tissues. We observed that exogenous acyl-GIP administration did not impact gastric emptying ($P > 0.05$; Fig. 2A), nor had any effects on plasma TG appearance ($P > 0.05$; Fig. 2B), findings which are supported by others (Regmi et al, 2024). Furthermore, we found no differences in insulin, c-peptide, or glucagon were observed between acyl-GIP and PBS-treated groups; however, active GLP-1 plasma concentrations were lower in acyl-GIP-treated animals during the first 30 min of the LTT (Fig. 2C–F). We assume this blunted response of active GLP-1 are associated with GIP being present in the circulation.

Overall, our findings show that GIP-mediated improvements in lipid tolerance are independent of lipid appearance from the gut to the circulation and suggest that lower lipid excursions are likely due to increased lipid clearance from the circulation into peripheral tissues. Therefore, we assessed how GIP treatment affects the uptake of circulating lipids during an oral lipid challenge by spiking olive oil with $^3$H-palmitic acid and sampling mice 2 h following oral lipid gavage. We discovered that GIP-treated mice increased BAT fatty acid uptake ~8% compared to PBS-treated mice ($P = 0.028$; Fig. 2G), with all other tissues showing no differences between PBS and GIP treatment ($P > 0.05$; Figs. 2G and EV2). While others have found GIP facilitates increased fatty acid uptake into primary white adipocytes (Killion et al, 2020) and eWAT following an oral lipid challenge (Regmi et al, 2024) in mice, we did not observe any differences in fatty acid uptake into WAT depots. This may be due to specific experimental protocols of administration of GIP used between studies. Nonetheless, our findings demonstrate that despite this lower dose of GIP, BAT increased fatty acid uptake during an oral lipid challenge.

Increased BAT LPL activity has been associated with enhanced lipid clearance from the circulation into BAT (Bartelt et al, 2011; Singh et al, 2018). To explore how GIP may be increasing BAT fatty acid uptake, we sampled obese mice 2 h following an injection of GIP or PBS, collected BAT, and measured LPL activity. We found that GIP-treated mice exhibited higher BAT LPL activity by ~21% compared to PBS-treated mice ($P = 0.048$; Fig. 2H). We also measured a trend to increase LPL activity in differentiated immortalized murine BAT cells with 1 h GIP treatment (10 and 50 nM), indicating that this might be a cell-autonomous effect of GIP on BAT (Fig. EV2Q). We also wanted to identify if this GIP-mediated increase in BAT lipid uptake during an LTT resulted in subsequent increases in whole-body fatty acid oxidation using indirect calorimetry; however, GIP administration during an LTT had no effect on whole-body energy expenditure or substrate oxidation (Fig. 3A–C). The GIPR signaling mechanisms of increased LPL activity have yet to be characterized in BAT. Previous work has shown that in murine-derived differentiated pre-adipocyte 3T3-L1 cells, GIP exposure increased LPL activity in a dose-dependent manner (Eckel et al, 1979; Kim et al, 2007).

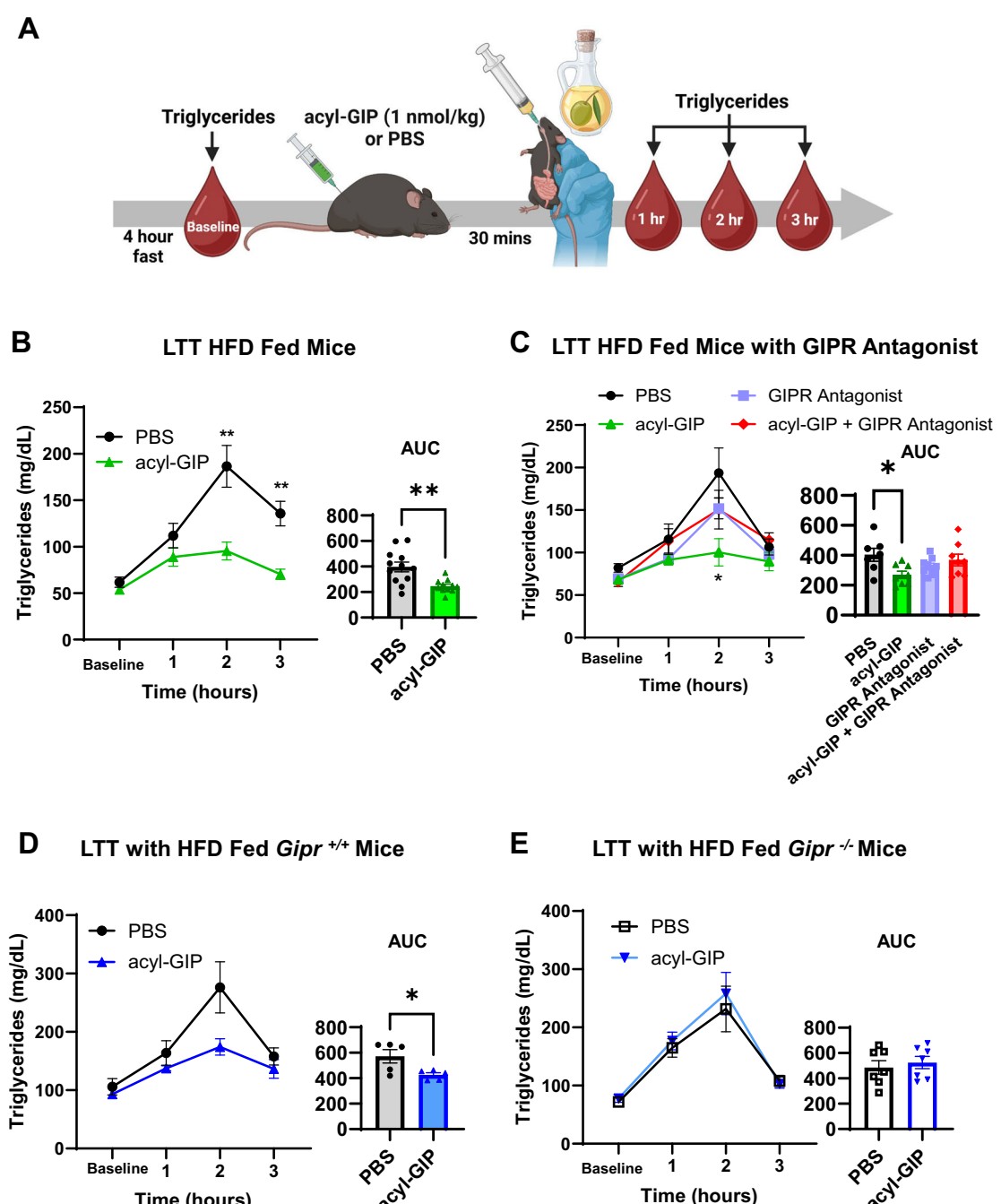

**Figure 1. Acute GIPR agonism improves lipid tolerance in diet-induced obesogenic (DIO) male mice.**

(A) Schematic for lipid tolerance test (LTT) protocol. Acute GIP administration (B): lowers plasma triglyceride excursions in obese mice (22-weeks old, PBS: $n = 12$, acyl-GIP: $n = 11$). (C) The effects of a GIPR antagonist (1500 nmol/kg) abolished GIPR agonist improvements of lipid tolerance in obese mice (26-week-old, PBS: $n = 7$, acyl-GIP: $n = 8$, GIPR antagonist: $n = 9$, acyl-GIP + GIPR antagonist: $n = 8$). The effects of acyl-GIP on 26-week-old obese (D) whole-body *Gipr* wild-type (*Gipr*[+/+]; $n = 5$ for PBS and acyl-GIP) and (E) whole-body *Gipr* knockout mice (*Gipr*[-/-]; $n = 7$ for PBS and acyl-GIP). Experiments were repeated minimum two times with each animal receiving either PBS or GIP in a cross-over design study, and data were pooled with two cohorts of animals. Two-way Repeated Measures ANOVAs were performed assessing the effects of treatment and time, with a Šidák's multiple comparisons test to compare between treatments at a given timepoint. Student unpaired *T* test and one-way ANOVA with a Dunnett's post hoc test were used to assess AUCs (area under the curve). *$P < 0.5$; **$P < 0.01$; ***$P < 0.001$; ****$P < 0.0001$. Data are presented as mean ± SEM. For exact *P* values, please refer to Dataset EV1. Source data are available online for this figure.

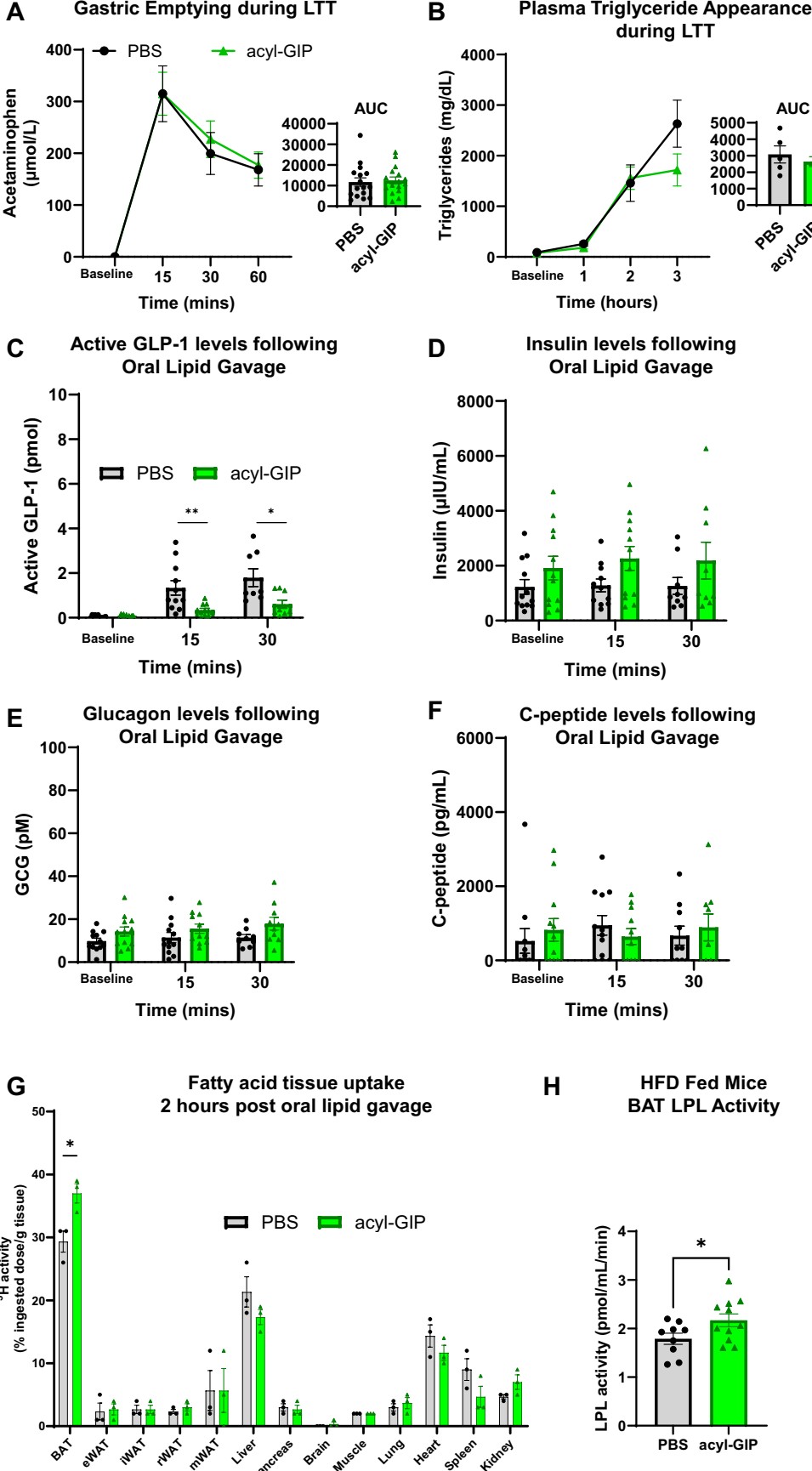

A **Gastric Emptying during LTT**

B **Plasma Triglyceride Appearance during LTT**

C **Active GLP-1 levels following Oral Lipid Gavage**

D **Insulin levels following Oral Lipid Gavage**

E **Glucagon levels following Oral Lipid Gavage**

F **C-peptide levels following Oral Lipid Gavage**

G **Fatty acid tissue uptake 2 hours post oral lipid gavage**

H **HFD Fed Mice BAT LPL Activity**

Figure 2.    Acute acyl-GIP administration increases plasma lipid clearance into BAT, rather than delay plasma lipid appearance, during an oral LTT in obese male mice.

Effects of exogenous acyl-GIP (1 nmol/kg) on (A) gastric emptying, determined by quantifying plasma appearance of acetaminophen (24-weeks old, $n = 16$ PBS and GIP).
(B) Plasma triglyceride appearance using lipoprotein lipase (LPL) inhibitor Polaxomer 407 (32 weeks old, $n = 5$ PBS and acyl-GIP) (C) Active GLP-1 plasma levels (26-weeks old, $n = 8$–12 PBS and acyl-GIP). (D) Plasma insulin levels (26-week-old, $n = 8$–12 PBS and acyl-GIP). (E) Plasma glucagon levels (26-week-old, $n = 8$–12 PBS and acyl-GIP). (F) C-peptide levels (26-week-old, $n = 8$–12 PBS and acyl-GIP). (G) Fatty acid distribution 2 h post oral lipid gavage ($n = 3$ for PBS and acyl-GIP). (H) BAT LPL activity 2 h following acyl-GIP administration in obese 30-week-old mice ($n = 9$ for PBS and $n = 11$ acyl-GIP). Experiments were repeated minimum two times with each animal receiving either PBS or GIP in a cross-over design study, and the data were pooled with 2 cohorts of animals. Except for the data described in (B, G), as the experiments were terminal. Student unpaired $T$ tests were used to compare PBS and acyl-GIP treatment groups and to assess AUCs (area under the curves). *$P < 0.5$; **$P < 0.01$. Data are presented as mean ± SEM. eWAT epididymal white adipose tissue, iWAT inguinal white adipose tissue, rWAT retroperitoneal white adipose tissue, mWAT mesenteric white adipose tissue. For exact $P$ values, please refer to Dataset EV1. Source data are available online for this figure.

Similarly, chronic elevation of GIP in rats resulted in increased LPL activity and TG accumulation in eWAT (Kim et al, 2007). Mechanistically identified in murine-derived differentiated pre-adipocytes, GIP increases protein kinase B (PKB) phosphorylation, leading to decreases in liver kinase B1 (LKB1) and 5'-adenosine monophosphate-activated protein kinase (AMPK) phosphorylation, causing increased LPL activation and subsequent increases in TG storage (Kim et al, 2007, 2010). Presumably, these cellular signaling mechanisms are similar in BAT, but less is known about this in an in vivo system.

Given our findings that exogenous acyl-GIP-induced reductions in lipid excursions during LTTs may be mediated through increases in BAT fatty acid uptake and LPL activity, we assessed how BAT GIPR signaling contributes to whole-body lipid tolerance, energy expenditure, and fatty acid oxidation using obese $Gipr^{BAT-/-}$ ($Gipr$ knockout) and littermate controls ($Gipr^{BAT+/+}$; Fig. 4A; for BAT $Gipr$ expression see Fig. EV3A). Consistent with our previous findings, we found that GIP administration decreased lipid excursions ~28% in $Gipr^{BAT+/+}$ mice compared to PBS-treated mice ($P = 0.027$; Fig. 4B); however, GIP had no effect on lipid tolerance in $Gipr^{BAT-/-}$ mice ($P = 0.8858$; Fig. 4C). These results mirrored our observations in whole-body GIPR knockout mice (Fig. 1J,K), suggesting that BAT GIPR signaling is a key driver in lowering lipid excursions during oral lipid challenges.

Previous work has demonstrated that GIP administration shifts substrate oxidation to favor lipids (Regmi et al, 2024; Zhang et al, 2021) independent of changes in energy expenditure (Zhang et al, 2021). This shift to preferentially oxidize fatty acids is appealing, particularly if it promotes fat-driven weight loss. Herein, mice that were placed in metabolic cages and acutely provided GIP displayed no differences in energy expenditure compared to PBS-treated mice, for both $Gipr^{BAT+/+}$ ($P > 0.05$; Fig. 4D) and $Gipr^{BAT-/-}$ mice ($P > 0.05$; Fig. 4E). Interestingly, RER was lower for $Gipr^{BAT+/+}$ mice for 6 h following GIP treatment compared to PBS-treated mice ($P < 0.01$; Fig. 4F), while these effects were absent in $Gipr^{BAT-/-}$ mice ($P = 0.8106$; Fig. 4G). Upon further elucidation of substrate use for energy expenditure, we found that for $Gipr^{BAT+/+}$ mice, whole-body fatty acid oxidation was ~11% higher for 6 h following GIP treatment compared to PBS-treated mice ($P < 0.01$; Fig. 4H), while these effects were absent in $Gipr^{BAT-/-}$ mice ($P = 0.559$; Fig. 4I). Unsurprisingly, the specific window of time where acyl-GIP increases fatty acid oxidation align with the 6–10 h half-life of the peptide (Mroz et al, 2019). Importantly, these increases in whole-body fatty acid oxidation were not associated with changes in food intake over a 24-h period following an overnight fast and then acyl-GIP administration ($P > 0.05$; Fig. 4J), unlike findings where others

have found GIP to decrease food intake in mice and subsequently increase preference for fatty acid oxidation (Regmi et al, 2024). Furthermore, when we administered either PBS or acyl-GIP (1 nmol/kg) treatment once daily for 3 days, neither $Gipr^{BAT+/+}$ or $Gipr^{BAT-/-}$ mice differed in cumulative food intake, and no changes in body mass were detected (Fig. EV3B–D). Therefore, our findings suggest that GIPR signaling in BAT is necessary for GIP-mediated increases in both lipid handling and whole-body fatty acid oxidation, all of which can occur independent of changes in food intake and body weight loss. Altogether, our findings are the first to suggest that BAT is a main driver for GIP-mediated increases in whole-body fatty acid oxidation in mice.

Previous work has highlighted how GIP-based medications increased genes involved with lipid catabolism and oxidation in BAT (Samms et al, 2021). Therefore, in addition to the increased fatty acid uptake into BAT, GIP may also signal for increased oxidation of these fatty acids. Alternatively, BAT GIPR signaling may not be linked to increases in fatty acid oxidation per se, but rather the availability of fatty acids to be oxidized. For example, in humans, GIP has been shown to increase AT blood flow (Asmar et al, 2010). Therefore, it may be possible that GIPR signaling in BAT promotes an increase in BAT blood flow. This increase in BAT blood flow, combined with an increase in BAT LPL activity, provides a scenario where GIP greatly increases the fatty acids available for BAT uptake, which may inherently contribute to an increase in fatty acid oxidation. A fruitful area of future research will be to uncover the mechanisms elucidating BAT GIP blood flow and BAT fatty acid oxidation.

Our data provides insight into the controversy of whether to agonize or antagonize GIPR for the treatment of metabolic disease (Campbell, 2020). Herein, we provide evidence that GIPR agonism, not antagonism, is necessary for acute GIP-mediated improvements in response to a bout of hypertriglyceridemia. Both pharmacological antagonism and genetic removal of GIPR (whole-body and BAT-specific) eliminated GIP-mediated improvements during an LTT. GIPR-antagonist-based medications have been demonstrated to lower fasting TG levels, though these findings may likely be correlated with decreases in food intake and body weight rather than mechanisms associated with increased TG clearance (Véniant et al, 2024). More work is needed to establish a deeper understanding of the impacts of GIPR-antagonists on the function of peripheral tissues.

While we observed GIP-mediated increases in fatty acid oxidation to be mainly driven by BAT, it is also possible that WAT is playing a contributing role. Despite WAT's low oxidative capacity (Frayn et al, 2008), work has shown that overexpression of

## A  Whole-body energy expenditure during an LTT

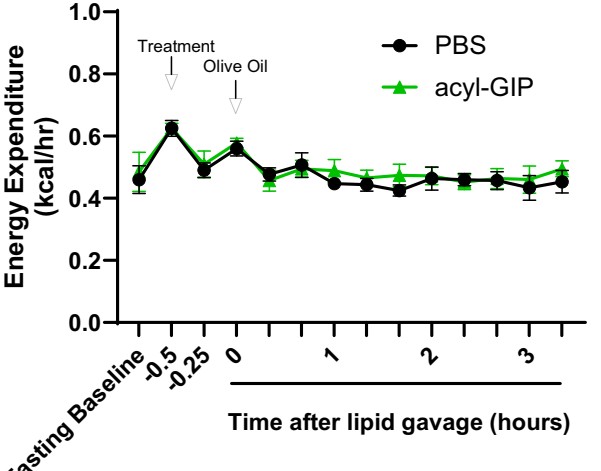

## B  RER during an LTT

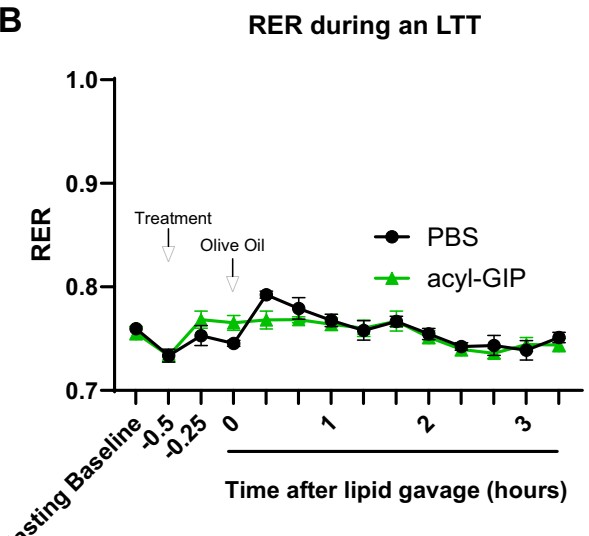

## C  Fatty acid oxidation during an LTT

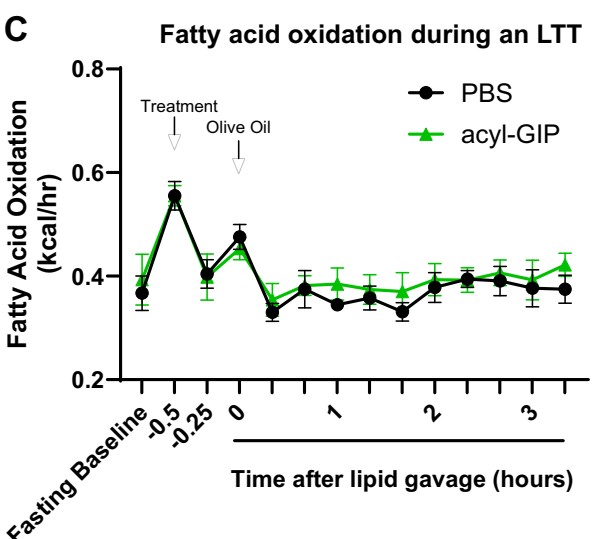

Figure 3.  Measures of energy expenditure and substrate oxidation during a LTT in 22-week-old obese male mice.

During an LTT, GIP had no effect on energy expenditure (**A**), respiratory exchange ratio (RER; (**B**) or whole-body fatty acid oxidation (**C**) compared to PBS-treated mice. ($n = 6$ for PBS and GIP). Experiments were repeated minimum two times with each animal receiving either PBS or GIP in a cross-over design study, and data were pooled with two cohorts of animals. Two-way repeated measures ANOVAs were performed assessing effects of treatment and time. Data are presented as mean ± SEM. For exact $P$ values, please refer to Dataset EV1. Source data are available online for this figure.

GIPR in the WAT of mice increases futile cycling of calcium, leading to increases in energy expenditure and fatty acid oxidation (Yu et al, 2025). Furthermore, GIP has been shown to promote lipolysis in WAT during the fasted state and promote lipid uptake during the fed state (Regmi et al, 2024). In our study, we did not observe any differences in radiolabelled palmitic acid uptake into WAT depots (Fig. 2G); however, this does not necessarily reflect the contribution of WAT to the plasma lipid pool. Changes in WAT lipolysis via GIP may contribute to the lipids available in the circulation for BAT to uptake. Unfortunately, the GIP-mediated crosstalk between BAT and WAT is unknown. More work is needed to explore how GIPR-expressing tissues contribute to whole-body substrate oxidation and energy expenditure.

Despite the significant insights of this study, there are several limitations to consider. Firstly, while we were able to trace the tissue destination of an ingested radiolabelled palmitic acid, this was only a snapshot of what was occurring in the body 2 h post gavage. Teasing apart tissue-specific fatty acid oxidation kinetics would have only been possible if we cannulated mice and infused GIP and collected blood in conscious mice throughout the 2 h period; however, this technique is extremely challenging and stress-inducing to the animals (Lyons and McClelland, 2024). However, with the development of fatty acid tracing through compound-specific isotope analysis, future studies can further elucidate fatty acid uptake and oxidation kinetics (Klievik et al, 2023). Secondly, as GIP has been previously demonstrated to alter adipose tissue blood flow, it may be possible that increases in GIP-mediated BAT uptake may have been a result of increased blood flow to the tissue. Thirdly, it is important to note that our rodent model of diet-induced obesity represents a model of insulin resistance but not diabetes directly. Fourthly, given the sex-based differences in lipid metabolism (Palmisano et al, 2018; Shi et al, 2009), we cannot extend findings from this current study to female mice. Studying the effects of exogenous GIP treatment in female mice remains a rich area of research and will increase our understanding of sex-based differences of GIPR-based medications. In addition, factors such as the half-life of the GIP, BAT activity, and rates of metabolism may be different within and between mouse models and humans, impacting the translatability of these findings (Boer et al, 2021; Demetrius, 2005; Perlman, 2016; Porter et al, 2016). Lastly, all mice in this study were housed at room temperature, which has been demonstrated to be a mild cold stress for mice and increased BAT activity (Chen et al, 2022; Clayton and McCurdy, 2018; McKie et al, 2019). If mice are housed at a temperature within their thermoneutral zone (30 °C), BAT activity is greatly diminished and provides a greater translational relevance to humans, given that active BAT has been identified in humans (Prapaharan

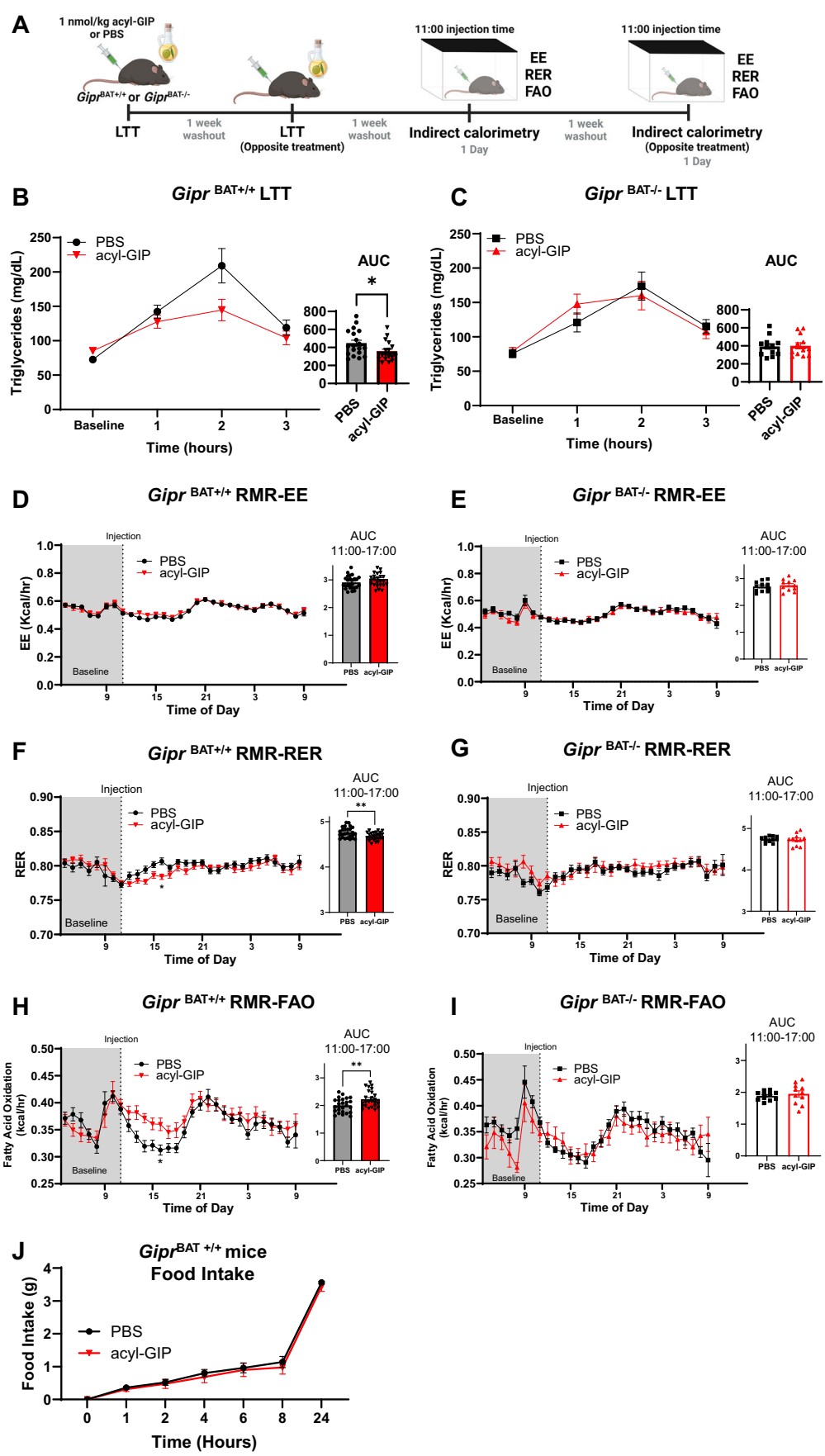

**Figure 4.   In obese male mice, BAT GIPR agonism is necessary for exogenous acute GIP-mediated improvements in lipid handling during lipid tolerance tests (LTT) and increases in whole-body fatty acid oxidation during routine metabolic rate (RMR).**

(A) Schematic for lipid tolerance and indirect calorimetry experiments. Acute acyl-GIP (1 nmol/kg) administration improves lipid tolerance in 24-week-old (B) *Gipr* expressing wild-type ($Gipr^{BAT+/+}$; $n = 19$ for PBS, $n = 21$ for acyl-GIP) mice, but not (C) *Gipr* knockout mice ($Gipr^{BAT-/-}$; $n = 12$ for PBS and acyl-GIP). During RMR conditions, acute acyl-GIP administration has no impact on energy expenditure (EE) in (D) $Gipr^{BAT+/+}$ ($n = 26$ for PBS and acyl-GIP) and (E) $Gipr^{BAT-/-}$ mice ($n = 11$ for PBS and acyl-GIP). Acute GIP administration decreases the respiratory exchange ratio (RER) within the first 6 h of treatment in (F) $Gipr^{BAT+/+}$ ($n = 26$ for PBS and acyl-GIP) but not (G) $Gipr^{BAT-/-}$ mice ($n = 11$ for PBS and acyl-GIP), demonstrating a subsequent increase in whole-body fatty acid oxidation (FAO) in (H) $Gipr^{BAT+/+}$ ($n = 26$ for PBS and acyl-GIP) and (I) $Gipr^{BAT-/-}$ mice ($n = 11$ for PBS and acyl-GIP). (J) Food intake following a 16 h overnight fast over a 24 h period in $Gipr^{BAT+/+}$ mice treated with GIP ($n = 8$) or vehicle control (PBS; $n = 5$). Experiments were repeated minimum two times with each animal receiving either PBS or GIP in a cross-over design study, and the data were pooled with three cohorts of animals. Two-way repeated measures ANOVAs and mixed effects analysis were performed assessing the effects of treatment and time, with a Šídák's multiple comparisons test to compare between treatments at a given timepoint. Student unpaired *t* test to compare AUCs, $*P < 0.05$, $**P < 0.01$. Data are presented as mean ± SEM. For exact *P* values, please refer to Dataset EV1. Source data are available online for this figure.

et al, 2024; Saito et al, 2009). Lastly, it is important to appreciate that in mice, BAT activity contributes a greater contribution to overall energy metabolism compared to humans (Perlman, 2016; Porter et al, 2016).

In summary, we highlight the importance of elucidating mechanisms of GIP on whole-body lipid handling and fatty acid oxidation. In doing so, we uncovered that GIPR signaling in BAT promotes BAT lipid uptake, which is a main driving factor for improved lipid handling and oxidation in GIP-treated obese mice. This increase in GIP-induced BAT lipid uptake was mechanistically associated with GIPR-linked increases in LPL activity. Our findings also allow us to appreciate how much BAT contributes to energy metabolism in preclinical studies with rodents. Given our findings regarding BAT's large contribution to whole-body fatty acid oxidation, previous work in rodents assessing how GIPR-based medications influence whole-body metabolism may have overlooked the influence of BAT. It may be that BAT is an important 'frontline sentinel' organ that contributes to lipid uptake under the influence of acute exogenous GIP administration. More molecular research is needed to identify how GIPR signaling in BAT improves lipid uptake. Future work studying GIPR-based medications in rodents should continue to consider BAT's role in energy metabolism and how this may impact our interpretations on the translational relevance to human metabolism.

## Methods

### Reagents and tools table

| Reagent/resource | Reference or source | Identifier or catalog number |
|---|---|---|
| **Experimental models** | | |
| Whole-body GIPR knockout ($Gipr^{-/-}$) and littermate controls ($Gipr^{+/+}$) | Miyawaki et al, 1999 | PMID: 12068290 |
| *Myf5*-Cre mice | The Jackson Laboratory | 007893 |
| Immortalized BAT Cells | Bruce Spiegelman, Uldry et al, 2006 | PMID: 16679291 |
| **Oligonucleotides and other sequence-based reagents** | | |
| *Gipr* | Mm01316344_m1 | Applied Biosystems |
| *Gapdh* | Mm99999915_g1 | Applied Biosystems |

| Reagent/resource | Reference or source | Identifier or catalog number |
|---|---|---|
| **Chemicals, enzymes, and other reagents** | | |
| Acyl-GIP | Novo Nordisk | |
| PBS | Wisent | 311-425-CL |
| GIPR antagonist | Novo Nordisk | |
| Glucose | Bioshop | GLU501 |
| Olive oil | MilliporeSigma | O1514 |
| Triglyceride Assay | Roche | 05171407190 |
| Acetaminophen | MilliporeSigma | A7085 |
| Acetaminophen assay kit | Sekisui Diagnostics | 506-30 |
| Poloxamer 407 LPL Inhibitor | MilliporeSigma | 16758 |
| Trasylol | MilliporeSigma | A6279 |
| EDTA | BioShop Canada Inc. | EDT002 |
| Diprotin A | MilliporeSigma | I9759 |
| Insulin, c-peptide, active GLP-1, and glucagon assay | Mesoscale Discovery | Cat# L45QXA-1 |
| $^3$H-palmitic acid | American Radiolabeled Chemicals | ARC1986 |
| Scintillation fluid | MilliporeSigma | UltimaGold LSC Cocktail L8286 |
| LPL assay kit | Abcam | ab204721 |
| TRIzol Reagent | Invitrogen | 15596026 |
| DNase I | ThermoScientific | EN0521 |
| Superscript III | Invitrogen | 56575 |
| Random Hexamers | Invitrogen | 58875 |
| TaqMan Gene Expression Master Mix and assays | ThermoScientific | 4444554 |
| Dulbecco's Modified Eagle Medium (DMEM); 4.5 g/l glucose, l-glutamine | Wisent | 319-015-CL |
| Fetal bovine serum (FBS) performance | Wisent | 98150 |
| Sodium Pyruvate | Wisent | 600-110-EL |
| Penicillin-streptomycin | Wisent | 450-201-EL |
| Dexamethasone | MilliporeSigma | D4902 |
| 3-isobutyl-1-methylxanthine (IBMX) | MilliporeSigma | I7378 |
| Insulin | MilliporeSigma | I7018 |

| Reagent/resource | Reference or source | Identifier or catalog number |
|---|---|---|
| Indomethacin | MilliporeSigma | I7378 |
| Triiodo-l-Thyronine (T3) | MilliporeSigma | T2877 |
| phenol-free DMEM | Wisent | 319-051-CL |
| EDTA-free protease inhibitor tablet | MilliporeSigma | 11836170001 |
| Phosphatase inhibitor 2 cocktail | Avantor | A80017-346 |
| Bradford reagent | Bio-Rad | 5000006 |
| **Software** | | |
| GraphPad Prism | GraphPad | version 9.5.0. |
| Biorender | Biorender | |
| **Other** | | |
| 60% High-fat diet | Research Diets | D12451i; Research Diets |
| Standard Regular Chow | Harlan Teklad | 2018 |
| Gluometer | Contour® NEXT EZ | |
| Meso Scale Diagnostics | Small Spot Assay system | L45QXA-1 |
| Scintillation Counter | Packard Bioscience Company | TRI-CARB 2900TR |
| Metabolic chambers | Columbus Instruments | Oxymax-CLAMS |
| Tissue Lyser | Qiagen | Tissue Lyser II system |
| qPCR machine | Thermo Fisher Scientific | QuantStudio System |

## Ethic approvals

All animal experiments were conducted in accordance with protocols approved by the Toronto Centre for Phenogenomics (TCP) Animal Care Committee (Animal Use Protocol # 25-0372H) and the University of Toronto Health Sciences Local Animal Care Committee (Animal Use Protocol # 20012708).

## Animals

Male C57BL/6J (The Jackson Laboratory, Bar Harbor, ME), whole-body GIPR knockout ($Gipr^{-/-}$) and littermate controls ($Gipr^{+/+}$) (Miyawaki et al, 1999), and GIPR knockout in BAT ($Gipr^{BAT-/-}$) and littermate controls ($Gipr^{BAT+/+}$) (Beaudry et al, 2019) were group housed (2–5 per cage) and maintained on a 12:12 light/dark cycle at room temperature (~23 °C) with food and water provided *ad libitum*. All mice were fed a standard rodent chow diet (RCD; 18% kcal from fat, 2018 Harlan Teklad, Mississauga, ON) unless designated to the obesogenic mouse group, where at 8 weeks of age mice were switched to a HFD (60% kcal from fat, D12451i, Research Diets, New Brunswick, NJ) for at least 12 weeks to induce obesity and insulin resistance. Mouse lean and fat mass were determined using EchoMRI nuclear magnetic resonance system (EchoMRI LLC, Houston, TX). $Gipr^{BAT-/-}$ and $Gipr^{BAT+/+}$ were generated by crossing hemizygous *Myf5*-Cre mice (007893, The Jackson Laboratory) with $Gipr^{Flox/Flox}$ mice that were maintained on

a C57BL/6J background (supplied by Dr. Daniel Drucker, Lunenfeld Tanenbaum Institute, Toronto, Canada). Despite *Myf5* being expressed in both skeletal muscle and BAT, skeletal muscle itself does not express GIPR, allowing this model to study the effects of *Gipr* in BAT. Wild-type, *Myf5*-Cre, and $Gipr^{Flox/Flox}$ mice were pooled as a single $Gipr^{BAT+/+}$ control group, as these mice were phenotypically similar in terms of body weight, body composition, and whole-body energy metabolism.

## Acyl-GIP and GIPR antagonist dosing and administration

Acyl-GIP (1 nmol/kg (formulation described in Mroz et al, 2019)) or vehicle control (PBS; 311-425-CL, Wisent Inc) was subcutaneously injected 30 min prior to the start of lipid, glucose, and insulin tolerance tests. In brief, palmitic acid was added to the N-terminus of a synthetic GIP analog to delay cleavage via dipeptidyl peptidase-4 inhibitor, extending the half-life from 4 to 7 min (native GIP) to 6–10 h (acyl-GIP). GIPR antagonist (1500 nmol/kg; (formulation and potency described in Yang et al, 2022) or PBS was subcutaneously injected 90 min prior to the start of tolerance tests. For whole-animal open-flow respirometry experiments, mice were briefly removed from the metabolic chamber/cage and were subcutaneously injected with 1 nmol/kg acyl-GIP or PBS at 11 AM.

## Glucose tolerance tests

All mice were fasted from 8 AM to 12 PM prior to tolerance tests. All blood measurements and samples were collected via the tail vein. For the glucose tolerance test, baseline blood glucose was measured using a glucometer (Contour® NEXT EZ). Following administrations of acyl-GIP and/or GIPR antagonist, mice were given an intraperitoneal injection of glucose (1.5 g/kg; GLU501, BioShop). Blood glucose was measured at 10, 20, 30, 60, 90, 120 min post glucose injection.

## Lipid tolerance tests

All mice were fasted from 8 AM to 12 PM prior to lipid tolerance tests (LTT). A baseline blood sample (~50 μL) was collected in a heparinized tube (Microvette CB 300 Lithium heparin, Starstedt), followed by agonist, antagonist, and/or vehicle control administration. Mice were then orally administered with 200 μL of olive oil (O1514, MilliporeSigma) using a 21-gauge gavage needle (Fine Science Tools). Blood samples were then collected at 1-, 2- and 3-h post gavage and kept on ice. Samples were spun at 10,000 × *g* at 4 °C for 5 min, and plasma was separated and stored at −80 °C until further analysis. Plasma TGs were assessed using the TG assay kit (05171407190, Roche Diagnostics).

## Gastric emptying

Using similar protocols of the LTT, acetaminophen (A7085 MilliporeSigma) was dissolved in the olive oil (3 mg/200 μL) and given to a subset of mice. Blood was collected after 4 h of fasting (baseline), and 15-, 30- and 60-min following gavage. Plasma acetaminophen was assessed using an acetaminophen assay kit (506-30, Sekisui Diagnostics).

## Plasma TG appearance

Plasma TG appearance was assessed in a subset of mice via intraperitoneal administration of Poloxamer 407 (P407; an LPL inhibitor, 1 g/kg body weight of P407); 16758 MilliporeSigma), dissolved in saline, followed immediately by olive oil gavage. Blood was collected from the tail vein after 4 h of fasting (baseline), and then at 1-, 2-, and 3-h post gavage to measure TG levels. As TGs continue to accumulate in the circulation after the P407 injection, mice in these experiments were sacrificed after blood collection was complete.

## Hormone analysis during LTT

Following LTT protocols, blood was collected at 4 h fasting baseline, 15- and 30-min post oral gavage of olive oil; in tubes containing TED (5000 kIU/ml), Trasylol (A6279, MilliporeSigma), 32 mM EDTA (EDT002, BioShop Canada Inc.), and 0.01 mM Diprotin A (I9759, MilliporeSigma)), which made up ~10% of the blood volume (~5 µL). Insulin, c-peptide, active GLP-1, and glucagon levels were assessed using Meso Scale Diagnostics (Cat# L45QXA-1, small spot assay system).

## Peripheral tissue fatty acid uptake after oral lipid challenge

A subset of mice underwent an LTT; however, each mouse received 50 µCi of $^3$H-palmitic acid (ARC1986, American Radiolabeled Chemicals) dissolved in 200 µL of olive oil. Two hours post-gavage, mice were anesthetized with isoflurane, and epididymal white adipose tissue (eWAT) and inguinal white adipose tissue (iWAT) were harvested, followed by a 5-min perfusion of cold PBS. eWAT and iWAT were removed prior to the perfusion to avoid blood contamination. After perfusion, the stomach, stomach contents, lungs, heart, kidney, small intestine, retroperitoneal white adipose tissue (rWAT), pancreas, spleen, muscle, brain, liver, and BAT were collected. Hard tissues were pulverized and soft tissues were homogenized by glass, and lipids were extracted using the Folch technique (Folch et al, 1957). The organic phase, representing radioactive lipids deposited into tissues, was separated, dried under $N_2$, and reconstituted in 5 ml of scintillation fluid (UltimaGold LSC Cocktail L8286, MilliporeSigma), and $^3$H radioactivity was measured by using a scintillation counter (TRI-CARB 2900TR, Packard BioScience Company). To determine the percentage of ingested isotope in an animal, we first subtracted the activity of the stomach, stomach contents, and small intestine from the total administered activity. Then, we divided the $^3$H activity of the extracted tissue (nCi/g) by the total amount ingested by the animal (nCi) to determine the percentage of $^3$H-palmitic activity within a tissue (% ingested dose/g organ).

## BAT LPL activity

BAT LPL activity was quantified using an LPL assay kit (ab204721; Abcam). Samples were prepared following kit instructions with a few modifications. Frozen BATs were weighed and homogenized in ice-cold PBS (0.25 mg tissue/µL PBS) and were then centrifuged at $10,000 \times g$ for 10 min at 4 °C. In total, 20 µL of the supernatant was loaded into the assay wells.

## Whole-animal open flow respirometry and food intake

To measure daily energy expenditure (EE) and RER, mice that were fed a HFD for at least 12 weeks were placed into metabolic chambers (Oxymax-CLAMS, Columbus Instruments) and were given 2 days to acclimatize. On the 2nd day, mice were treated with either acyl-GIP or PBS at 11:00 AM, and then oxygen consumption, carbon dioxide production, RER, and energy expenditure (kcal/hr) were recorded for 24 h. Data were averaged every hour for the entirety of the experiment. Whole-body fatty acid oxidation rates were calculated using the following equation: fatty acid oxidation (kcal/h) = energy expenditure × ((1 − RER)/0.3). Food intake was measured 24 h following treatment while mice were in the metabolic cages. Ad libitum food intake was also assessed in a separate group of mice after an overnight fast and then an injection of acyl-GIP (1 nmol/kg) for 1-, 2-, 4-, 6-, 8-, and 24-h.

## RNA isolation, cDNA, and qPCR

RNA isolation, cDNA, and qPCR were completed using similar methods as previously published work (Beaudry et al, 2019). Frozen BAT was homogenized in TRIzol Reagent (TRIzol Reagent 15596026, Invitrogen) using a TissueLyser II system (Qiagen, Germantown, MD), and total RNA was extracted using the manufacturer's protocol. First-strand cDNA was synthesized from DNase I-treated total RNA (2 µg, EN0521, Thermo Fisher Scientific) using SuperScript III (56575, Invitrogen) and random hexamers (58875, Invitrogen). A QuantStudio System and TaqMan Gene Expression Master Mix and Assays (4444554, Thermo Fisher Scientific) were used to quantify gene expression. The following primer/probes and their catalog numbers were purchased from Thermo Fisher Scientific: *Gipr* (Mm01316344_m1), while expression levels were normalized to *Gapdh* (Mm99999915_g1), and gene expression was analyzed by $2^{-\Delta\Delta Ct}$ method. Final values were made relative to regular chow diet-fed mice brown adipose tissue *Gipr* levels.

## Immortalized brown adipocyte lipoprotein lipase activity

Similar methods were previously described in Beaudry et al, 2019 where immortalized BAT cells from Dr. Bruce Spiegelman's laboratory (Department of Cancer Biology, Dana-Farber Cancer Institute, Boston, MA) were cultured on collagen-coated 6-well plates at a seeding density of 120,000 cells per well at 37 °C, 5% $CO_2$, in growth media (Dulbecco's Modified Eagle Medium (DMEM); 4.5 g/l glucose, l-glutamine, Wisent, cat# 319-015-CL) supplemented with 20% fetal bovine serum (FBS) performance (Wisent, cat# 98150), 1% sodium pyruvate (100 mM, Wisent, cat# 600-110-EL), and 1% penicillin-streptomycin (Wisent, cat# 450-201-EL). As cells were confluent, growth media was removed and induction media was added to the cells (DMEM supplemented with 10% FBS, 1% sodium pyruvate, 1% penicillin-streptomycin, 500 nM dexamethasone (MilliporeSigma, cat# D4902), 0.5 mM 3-isobutyl-1-methylxanthine (IBMX; MilliporeSigma, I7018), 125 mM indomethacin (MilliporeSigma, cat# I7378), 1.7 µM insulin (MilliporeSigma, cat# I9278), and 1 nM T3 (Triiodo-l-Thyronine, MilliporeSigma, cat# T2877)). After a 48-h induction period, maintenance media (DMEM supplemented with 10% FBS, 1% sodium pyruvate, 1% penicillin-streptomycin, 1.7 µM insulin, and

1 nM T3) replaced the induction media and were changed every 2 days until cells were fully differentiated by day 8. On day 8, experimental media (phenol-free DMEM; Wisent, cat # 319-051-CL, supplemented with 3.5% FBS, 1% penicillin-streptomycin) replaced maintenance media for 1 h, followed by the further addition of either acyl-GIP (10 or 50 nM) or vehicle control (PBS) for 2 h. Protein was extracted using lysis buffer (150 mM NaCl, 20 mM Trizma base, 2 mM EDTA, 0.5% Triton X-100, EDTA-free protease inhibitor tablet (MilliporeSigma, cat# 11836170001), and phosphatase inhibitor 2 cocktail (Avantor cat# A80017-346), centrifuged at 14,000 ×g for 15 min, and quantified using a Bradford assay (Bio-Rad, cat# 5000006). A total of 50 μg of protein was loaded per assay well for determining LPL activity using the LPL assay kit (ab204721; Abcam). Each data point represents an independent biological replicate.

## Statistics

All reported data are expressed as mean ± SEM. Two-way repeated measure ANOVAs and mixed effects analysis were used to analyze the interactions between treatment and time for tolerance tests, energy expenditure, RER, and whole-body fatty acid oxidation. Pairwise, Holm–Sidak post hoc tests were performed to assess significant interactions. Student $t$ tests were performed to compare differences in treatment groups for the area under the curves (AUC) and all other dependent variables. Two-way ANOVA was used to assess LPL activity, with genotype and treatment as factors, and a Tukey's post hoc test was used to identify any significant interactions. A Grubb's outlier test was performed to identify and eliminate outliers. A statistical significance value was set to $P < 0.05$. GraphPad Prism version 9.5.0. was used to perform all statistical analyses. GraphPad Prism and BioRender were used to generate figures.

## Data availability

This study includes no data deposited in external repositories.

The source data of this paper are collected in the following database record: biostudies:S-SCDT-10_1038-S44319-025-00582-7.

## Peer review information

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

## Acknowledgements

The authors would like to thank Zahra Shahabi, Drs. Jackie Koehler and Rola Hammoud for instructional and technical assistance for tolerance tests. We

would like to thank Drs. Erin Mulvihill and Carolyn Cummins for gastric emptying and plasma TG formation protocols, and Daniel Drucker for the donation of *Gipr*<sup>Flox/Flox</sup> mice. We also would like to thank the Scientific Director for Metabolic Research Dr. Bharath Mani and Senior Scientist in Receptor Pharmacology, Dr. Jacek Mokrosinski at Novo Nordisk for their coordination for in-kind contributions to use the experimental reagents to generate data for this article. JLB is supported in part by various funds, including a Canadian Foundation for Innovation (CFI)-Ontario Research Fund (ORF) and CFI- John R. Evans Leaders Fund (JELF), CIHR Project Grant, NSERC-Discovery, BBDC; Novo Nordisk-New Investigator Award, and Drucker Family Innovation Fund. SAL was supported by NSERC PDF. MBSL was supported by CIHR (CGS-M). SAL and MP were supported by the BBDC D. H. Gales Family Charitable Foundation Postdoctoral Fellowship. ZG was supported by the China Scholarship Council. ARB was supported by BBDC summer student scholarships. SZ was supported by the BBDC graduate student scholarship. JRP was supported by the Undergraduate Research Opportunity Program. AHM was supported by NSERC, Canadian Foundation for Innovation (CFI)-Ontario Research Fund (ORF) and CFI- John R. Evans Leaders Fund (JELF). RPB was supported by NSERC and a Canada Research Chair in Brain Lipid Metabolism. JEC was supported by funding from the NIH/NIDDK (DK046492, DK125353, DK123075, and DK141090).

## Author contributions

**Sulayman A Lyons**: Data curation; Formal analysis; Investigation; Methodology; Writing—original draft; Writing—review and editing. **Micah B S Lea**: Data curation; Formal analysis; Investigation; Methodology; Writing—original draft; Writing—review and editing. **Mihir Parikh**: Data curation; Formal analysis; Investigation; Methodology; Writing—original draft; Writing—review and editing. **Zhengzhang Guo**: Data curation; Formal analysis; Investigation; Methodology; Writing—review and editing. **Samrin Kagdi**: Data curation; Formal analysis; Investigation; Methodology; Writing—review and editing. **Abigail R Bisnauth**: Data curation; Formal analysis; Investigation; Methodology; Writing—review and editing. **Jonathan R Pitino**: Data curation; Formal analysis; Methodology; Writing—review and editing. **Sabrina Ziai**: Data curation; Formal analysis; Methodology; Writing—review and editing. **Negar Mir**: Data curation; Formal analysis; Methodology. **Aidan D Tyrrell**: Data curation; Methodology; Writing—review and editing. **Yan Fu**: Data curation; Methodology. **Chuck T Chen**: Data curation; Formal analysis; Methodology; Writing—review and editing. **Adam H Metherel**: Data curation; Formal analysis; Investigation; Methodology; Writing—review and editing. **Richard P Bazinet**: Data curation; Formal analysis; Supervision; Writing—review and editing. **Bin Yang**: Formal analysis; Methodology. **Patrick J Knerr**: Formal analysis; Methodology; Writing—review and editing. **Jonathan D Douros**: Formal analysis; Writing—review and editing. **Jonathan E Campbell**: Conceptualization; Formal analysis; Investigation; Writing—review and editing. **Jacqueline L Beaudry**: Conceptualization; Formal analysis; Supervision; Funding acquisition; Methodology; Writing—original draft; Project administration; Writing—review and editing.

Source data underlying figure panels in this paper may have individual authorship assigned. Where available, figure panel/source data authorship is listed in the following database record: biostudies:S-SCDT-10_1038-S44319-025-00582-7.

## Disclosure and competing interests statement

PJK, BY, and JDD are former employees and shareholders of Novo Nordisk. JEC receives funding for basic science from Eli Lilly, Novo Nordisk, Merck, Structure Therapeutics, Fractyl Health, and Prostasis. JEC has served as an advisor in the past 12 months to Arrowhead Pharma, Boehringer Ingelheim, Protagonist, Prostasis, Septerna, and Structure Therapeutics. AHM is on the Board of Directors of the International Society for the Study of Fatty Acids and Lipids, is a Science Advisor for Benexia and Natures Crops International, and is a co-applicant on a joint government/industry-funded research grant with Natures Crops International.

# Expanded View Figures

**Figure EV1. Assessment of GIPR agonism and GIPR antagonism on glucose and lipid tolerance in male mice.**

Comparison in (**A**) body weight, (**B**) body composition and (**C**) 4-h fasting blood glucose levels between lean (regular chow diet; RCD) and obese (60% high-fat diet; HFD) 20-week-old male C57BL/6 mice ($n = 15$–16 for RCD; $n = 32$–36 for HFD). Determination of the efficacy of acyl-GIP during a lipid tolerance test (LTT; (**D**) and intraperitoneal glucose tolerance test (IPGTT; (**E**) Acute acyl-GIP (1 nmol/kg) administration lowers glucose excursions during an intraperitoneal glucose tolerance test (IPGTT) in (**F**) lean mice (20-weeks old, $n = 8$ for PBS and acyl-GIP) and (**G**) obese mice (20-weeks old, $n = 8$ for PBS and acyl-GIP). The impact on glucose tolerance when both acyl-GIP (1 nmol/kg) and a GIPR antagonist (1500 nmol/kg) is administered in (**H**) 20–30-week-old obese male mice ($n = 3$–7 per treatment group). The effects of acyl-GIP on 20-week-old lean (**I**) whole-body *Gipr* wild-type (*Gipr*$^{+/+}$; PBS and acyl-GIP, $n = 4$) and (**J**) whole-body *Gipr* knockout mice (*Gipr*$^{-/-}$; PBS and acyl-GIP, $n = 5$). (**K**) A comparison of various adipose tissue depot *Gipr* mRNA expression between RCD and HFD-fed mice. All values made relative to RCD BAT *Gipr* levels ($n = 6$–7 for RCD and HFD mice). Experiments were repeated minimum 2 times with each animal receiving either PBS or GIP in a cross-over design study and data was pooled with 2 cohorts of animals. Except for data described in K, as the experiments were terminal. Two-way Repeated Measures ANOVAs or linear mixed effects analysis were performed assessing effects of treatment and time, with a Šídák's multiple comparisons test to compare between treatments at a given timepoint. One-way ANOVAs with a Dunnett's post hoc test were used to assess AUCs (area under the curve) compared to PBS group. Student unpaired $T$ tests were used to compare PBS and GIP treatment groups, and to compare RCD and HFD-fed mice. *$P < 0.5$; **$P < 0.01$; ***$P < 0.001$; ****$P < 0.0001$. $ Significant difference between 1 and 10 nmol/kg groups and PBS ($P < 0.05$). # Significant differences between 10 nmol/kg and PBS ($P < 0.05$). Data are presented as mean ± SEM. BAT, brown adipose tissue. iWAT, inguinal white adipose tissue. eWAT, epidymal white adipose tissue. rWAT, retroperitoneal white adipose tissue. For exact $P$ values, please refer to Dataset EV1. Source data are available online for this figure.

▶

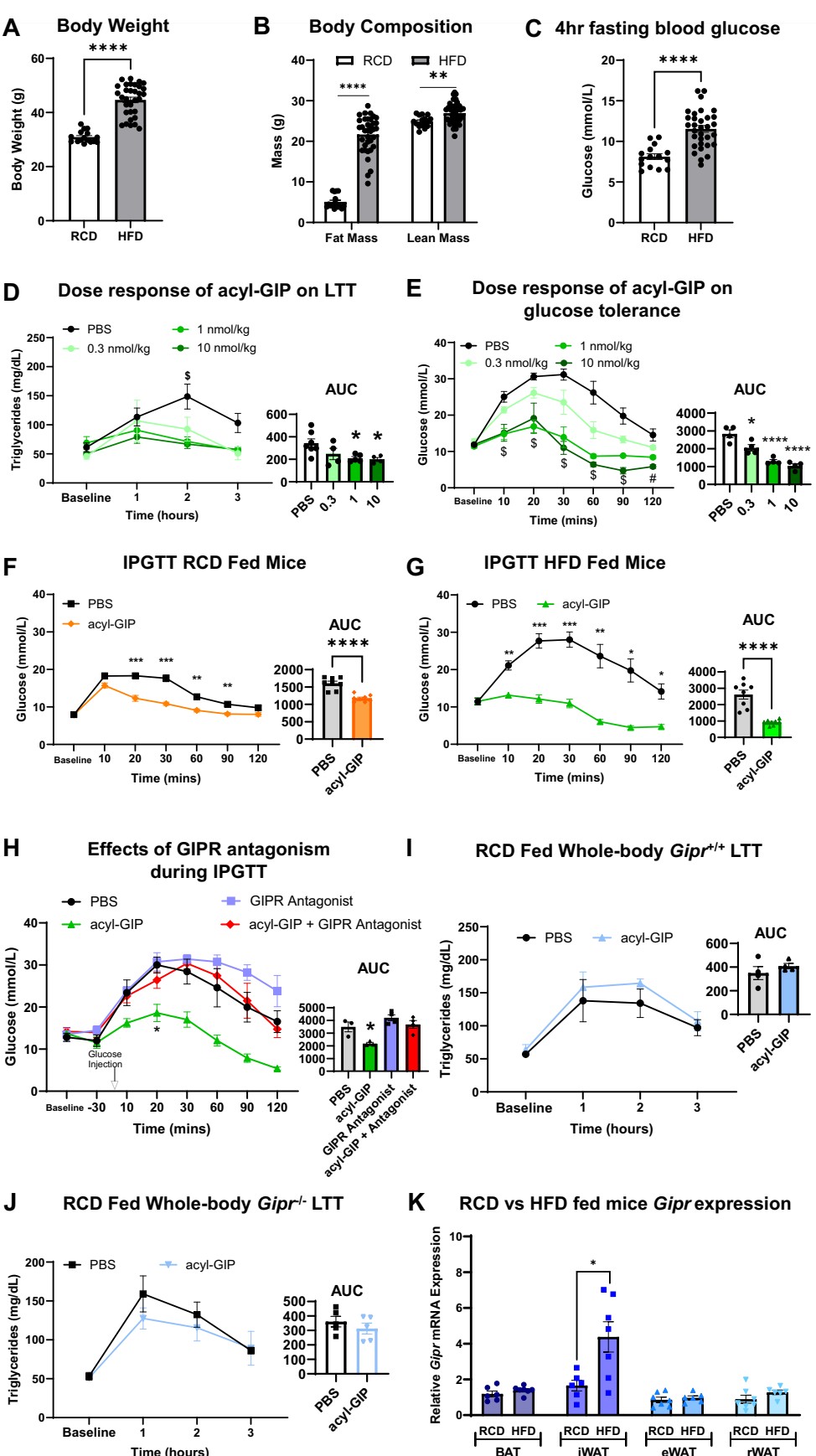

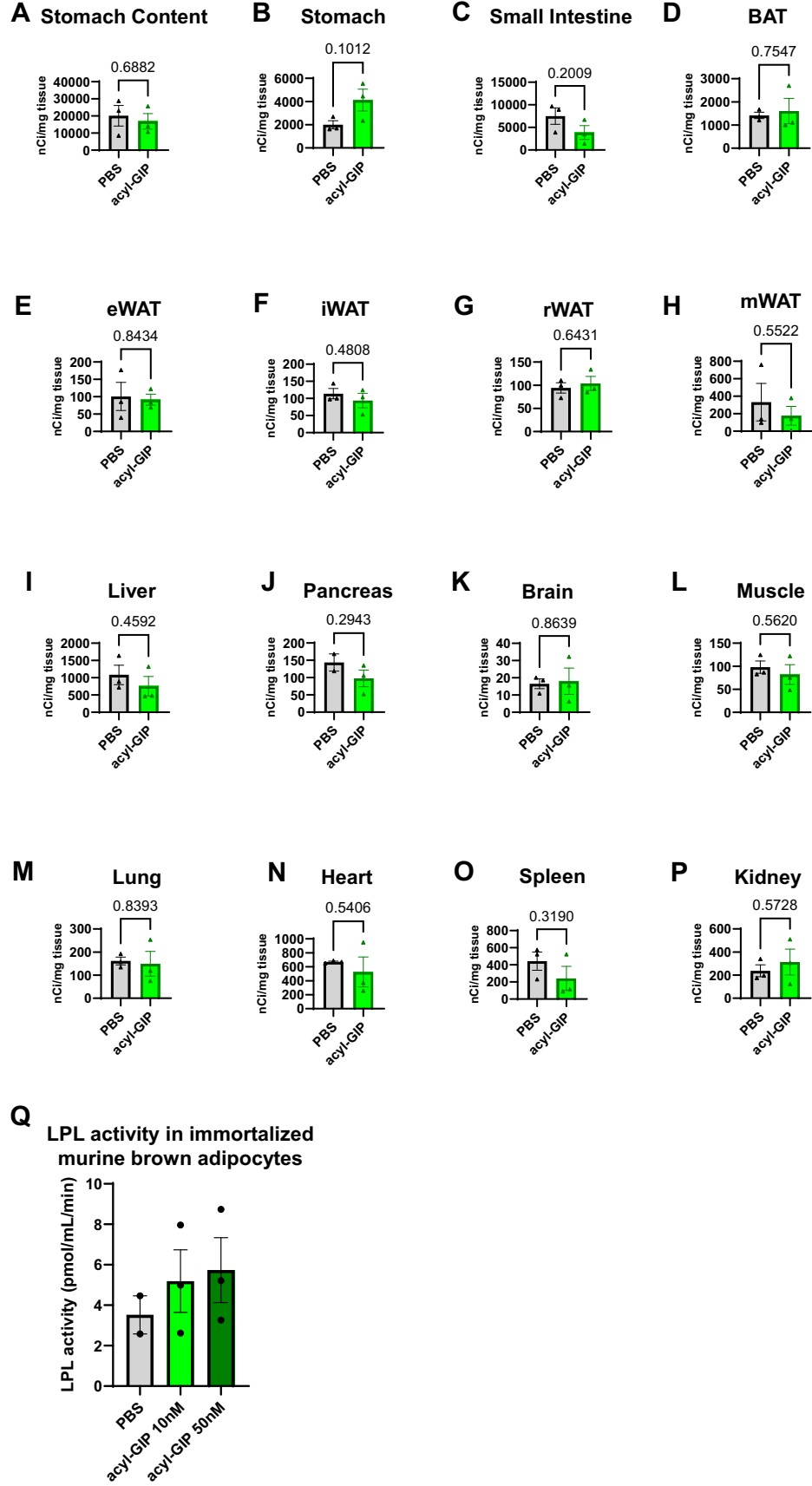

◀ **Figure EV2. Values for ³H-palmitic acid uptake 2 h following an oral lipid challenge in male obese mice fed a 60% HFD for 12 weeks, and lipoprotein lipase (LPL) activity of day 8 differentiated murine brown adipocytes.**

Lipid phase ³H-activity relative to tissue mass in a variety of mouse tissues used to calculate relative % distribution for Fig. 2G. $n = 3$ for PBS and acyl-GIP. Experiments were completed in 1 cohort of mice as these experiments were terminal (**A–P**). Student unpaired $T$ tests were used to compare PBS and GIP treatment groups. (**Q**) 8 day differentiated immortalized murine brown adipose tissue cells exposed to various concentrations of acyl-GIP to assess lipoprotein lipase (LPL) activity. Each data point represents 1 biological replicate ($n = 2$ for pbs, $n = 3$ for 10 and 50 nM acyl-GIP). One-way ANOVA with a Dunnett's post hoc test was used to compare PBS to all GIP treatments. Data are presented as mean ± SEM. Source data are available online for this figure.

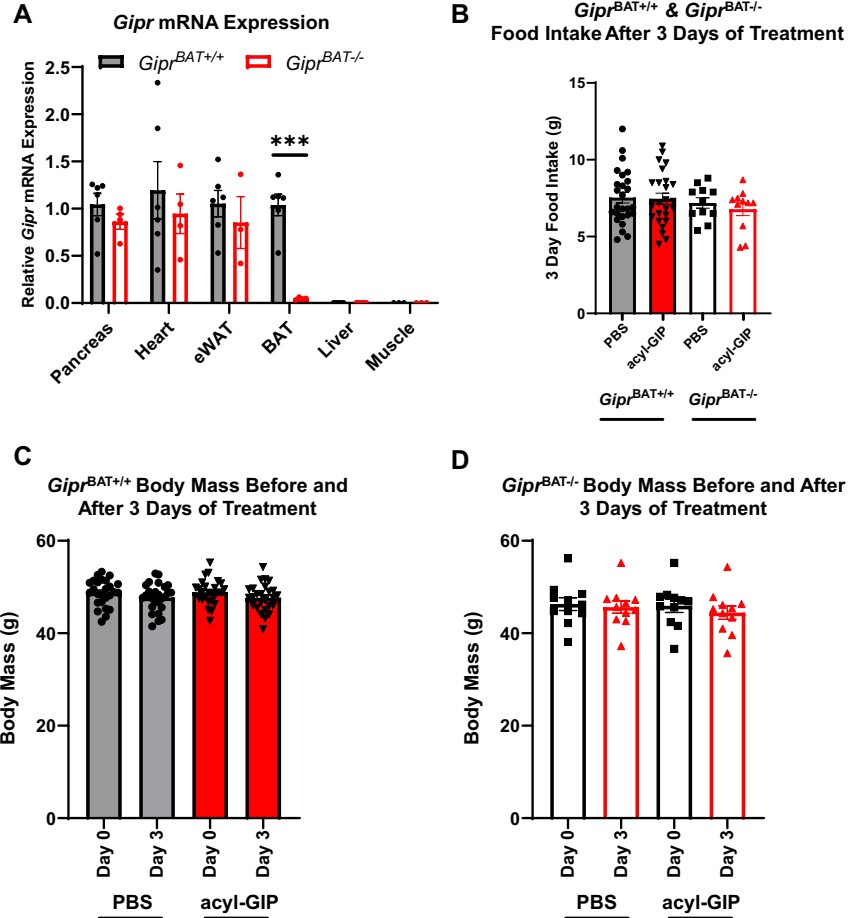

**Figure EV3. BAT specific GIPR knockout mouse model.**

(A) *Gipr* gene expression in a variety of tissues, confirming tissue specific knockout in the BAT (*Gipr*BAT +/+; n = 3–6, *Gipr*BAT -/-; n = 3–4). (B) After 3 consecutive days of PBS of acyl-GIP (1 nmol/kg) treatment, there was no change in the cumulative food intake of obese male *Gipr*BAT +/+ (n = 26) and *Gipr*BAT -/- (n = 11) mice. After 3 consecutive days of PBS of acyl-GIP (1 nmol/kg) treatment, no change in body mass was observed in (C) *Gipr*BAT +/+ or (D) *Gipr*BAT -/- obese mice (*Gipr*BAT +/+; n = 26, *Gipr*BAT -/-; n = 11). Experiments were repeated minimum 2 times with each animal receiving either PBS or GIP in a cross-over design study and data was pooled with 3 cohorts of animals. ***P < 0.001. Student unpaired *T* tests were used to compare genotypes, treatment groups, and time on treatment. Data presented as mean ± SEM. For exact *P* values, please refer to Dataset EV1. Source data are available online for this figure.

