## [Peer Review File · EMBO Reports]

Acute exogenous acyl-GIP treatment enhances lipid handling and fatty acid oxidation by involving brown fat.

Sulayman Lyons, Micah Lea, Mihir Parikh, Zhengzhang Guo, Samrin Kagdi, Abigail Bisnauth, Jonathan Pitino, Sabrina Ziai, Negar Mir, Aidan Tyrrell, Yan Fu, Chuck Chen, Adam Metherel, Richard Bazinet, Bin Yang, Patrick Knerr, Jonathan Douros, Jonathan Campbell, and Jacqueline Beaudry

Corresponding author(s): Jacqueline Beaudry (jacqueline.beaudry@utoronto.ca)

Review Timeline:

Transfer Date:	7th May 25
Editorial Decision:	9th May 25
Revision Received:	22nd May 25
Editorial Decision:	17th Jul 25
Revision Received:	8th Aug 25
Accepted:	30th Aug 25

Editor: Deniz Senyilmaz Tiebe

Transaction Report: This manuscript was transferred to EMBO reports following peer review at The EMBO Journal.

Referee #1:

This paper by Lyons et al. demonstrates that acyl-GIP improves lipid tolerance in obese male mice in a manner dependent on GIPR in Ucp1+ adipocytes.

This is a significant contribution to advancing our understanding of a medically relevant topic. The finding that brown adipose tissue (BAT) plays a metabolically protective role through pharmacological GIP is novel and underscores the need for further exploration of the therapeutic potential of brown fat. I have a few comments and suggestions that could strengthen this already compelling work.

1. Line 122-124: Could the ~8% increase in fatty acid (FA) uptake by BAT be converted to absolute amounts, accounting for BAT volume/abundance? This could provide a clearer understanding of FA uptake and clearance at the whole-body level. Additionally, it may be useful to compare the delta between 3H-palmitic acid tracing into BAT in wild-type (WT) and BAT-specific GIPR knockout (KO) mice.
2. The hypothesis that GIP alters blood flow is intriguing. This could be easily tested using FITC-dextran, with or without acyl-GIP, in WT mice, which would help rule in or out this potential mechanism.

Referee #2:

This paper shows that acute treatment with GIP increases LPL activity and lipid uptake in BAT of obese mice. The effect of GIP on lipid tolerance was absent in whole body or BAT-selective GIPR knockout animals. RER values were also decreased by GIP treatment without a detectable change in energy expenditure or food intake. Interestingly, the effect of GIP on lipid clearance was not observed in chow fed (lean) animals. The experiments are topical given the great interest in understanding effects of GIP on body weight and metabolism. The studies are also technically sound. However, the paper is quite descriptive and could be strengthened by providing some mechanistic information regarding how GIP signaling may modulate BAT LPL activity, and why this may be selective to high fat diet/obesity conditions.

Additional comments follow:

1. Related to the mechanism by which GIP treatment increases LPL activity in BAT- have the authors monitored the LKB1-AMPK pathway in the BAT (vs. white fat depots) in their

treatment paradigms?

2. Is there a change in GIPR expression in brown adipocytes or other BAT cell types from chow/lean vs. hfd/obese animals?
3. The effect on "whole body FAO" was driven by a decrease in the RER - this could reflect a switch towards fatty acid utilization in BAT due to increased lipid uptake. Did the authors assess if glucose uptake is lowered in the BAT following treatment?
4. It would be interesting to know if the lipid clearance effects of GIP are diminished under thermoneutral conditions.
5. The Myf5Cre used for BAT KO would also eliminate GIPR in muscle and some other tissues during development. This should be discussed.

Referee #3:

This manuscript "Acute exogenous acyl-GIP administration enhances whole-body lipid handling and fatty acid oxidation via brown adipose tissue in obese male mice" reports a positive effect of an acylated glucose-dependent insulinotropic polypeptide (acyl-GIP) on lipid handling and fatty acid oxidation in mice. They further use GIPR inhibitor and Knockout mice to demonstrate that the effect of acyl-GIP depends on GIPR specifically in the brown adipocytes. The results have strong translational implication as it shows that GIPR signaling in brown fat underly GIP-induced systemic improvements of lipid metabolism. However, the data presented do not sufficiently support this major conclusion.

The first and foremost concern the Myf5Cre that was used to drive Gipr knockout in the brown adipocytes. The Cre driver is known to drive gene deletion in skeletal muscle progenitors (and therefore all muscle cells, PMID: 17540178), as well as other cell types in addition to muscle and fat (PMID: 20848654). As the skeletal muscle accounts for 40% of body weight and plays a key role in glucose and fatty acid metabolism, the role of muscle-specific GIPR signaling should have been analyzed to support the conclusion.

The study overly relies on LTT data and a couple of related physiological assays, which are used to reflect the systemic response to lipid overload, but in depth molecular mechanistic analysis and function validation is lacking.

Minor Comments:

Methods often fall short of providing necessary information to others to repeat the work.

Some sections have only 2 sentences. For instance: What is fatty acid in the acyl-GIP, how it was formulated, and what is purpose of this modification?

Dear Jacqueline,

Thank you for transferring your manuscript to EMBO Reports, which was previously reviewed at The EMBO Journal.

Having read the manuscript and the referee reports, I would like to invite you to submit a revised manuscript to EMBO Reports as previously communicated in my EMBO Journal colleague Daniel's decision letter. In particular,

- Both referees #2 and #3 point out the expression of the Myf5Cre driver in the skeletal muscles in addition to BAT. You have already shared your arguments addressing this point with me in our follow-up exchange. Please include them into your point-by-point response to the referee concerns prior to submitting the revised manuscript.
- The effect of acyl-GIP treatment on the LKB1-AMPK pathway in BAT and WAT should be investigated (referee #2).
- Although specific comments of referees need to be addressed, elucidating the mechanism by which acyl-GIP regulates lipid handling is not required for publication in EMBO Reports. Similarly, please include your arguments you already shared with me addressing the concerns of referee #1 regarding the GIPR depletion mouse model used in the study.

Please address all referee concerns in a complete point-by-point response. Acceptance of the manuscript will depend on a positive outcome of a second round of review. It is EMBO reports policy to allow a single round of major experimental revision only and acceptance or rejection of the manuscript will therefore depend on the completeness of your responses included in the next, final version of the manuscript.

Please contact me if you have questions or comments regarding any of these points or the revision for further discussion (also by video chat).

Given these recommendations, we would like to invite you to revise your manuscript with the understanding that the referee concerns (as in their reports) must be fully addressed and their suggestions taken on board. Please address all referee concerns in a complete point-by-point response. Acceptance of the manuscript will depend on a positive outcome of a second round of review. It is EMBO reports policy to allow a single round of major experimental revision only and acceptance or rejection of the manuscript will therefore depend on the completeness of your responses included in the next, final version of the manuscript.

We realize that it is difficult to revise to a specific deadline. In the interest of protecting the conceptual advance provided by the work, we recommend a revision within 3 months. Please discuss the revision progress ahead of this time with me if you require more time to complete the revisions, or if you have questions or comments regarding the revision (also by video chat).

1. A data availability section providing access to data deposited in public databases is missing (where applicable).
2. Your manuscript contains statistics and error bars based on n=2. Please use scatter plots in these cases.

You can submit the revision either as a Scientific Report or as a Research Article. For Scientific Reports, the revised manuscript can contain up to 5 main figures and 5 Expanded View figures, and it should not exceed 27000 characters. If the revision leads to a manuscript with more than 5 main figures it will be published as a Research Article. In this case the Results and Discussion section should be separate. If a Scientific Report is submitted, these sections have to be combined. This will help to shorten the manuscript text by eliminating some redundancy that is inevitable when discussing the same experiments twice. In either case, all materials and methods should be included in the main manuscript file.

3) We replaced Supplementary Information with Expanded View (EV) Figures and Tables that are collapsible/expandable online. A maximum of 5 EV Figures can be typeset. EV Figures should be cited as 'Figure EV1, Figure EV2' etc... in the text and their respective legends should be included in the main text after the legends of regular figures.

4) a .docx formatted letter INCLUDING the reviewers' reports and your detailed point-by-point responses to their comments. As part of the EMBO publication's Transparent Editorial Process, EMBO reports publishes online a Review Process File (RPF) to accompany accepted manuscripts. This File will be published in conjunction with your paper and will include the referee reports, your point-by-point response and all pertinent correspondence relating to the manuscript.

<https://www.embopress.org/page/journal/14693178/authorguide#transparentprocess>

5) a complete author checklist, which you can download from our author guidelines

<https://www.embopress.org/page/journal/14693178/authorguide>. Please insert information in the checklist that is also reflected in the manuscript. The completed author checklist will also be part of the RPF.

6) Please note that all corresponding authors are required to supply an ORCID ID for their name upon submission of a revised manuscript (<<https://orcid.org/>>). Please find instructions on how to link your ORCID ID to your account in our manuscript tracking system in our Author guidelines

<<https://www.embopress.org/page/journal/14693178/authorguide#authorshipguidelines>>

7) Before submitting your revision, primary datasets produced in this study need to be deposited in an appropriate public database (see <https://www.embopress.org/page/journal/14693178/authorguide#datadeposition>). Please remember to provide a reviewer password if the datasets are not yet public. The accession numbers and database should be listed in a formal "Data Availability" section placed after Materials & Method (see also

<https://www.embopress.org/page/journal/14693178/authorguide#datadeposition>). Please note that the Data Availability Section is restricted to new primary data that are part of this study. * Note - All links should resolve to a page where the data can be accessed. *

Additional information on source data and instruction on how to label the files are available:

<https://www.embopress.org/page/journal/14693178/authorguide#sourcedata>

9) Our journal encourages inclusion of *data citations in the reference list* to directly cite datasets that were re-used and obtained from public databases. Data citations in the article text are distinct from normal bibliographical citations and should directly link to the database records from which the data can be accessed. In the main text, data citations are formatted as follows: "Data ref: Smith et al, 2001" or "Data ref: NCBI Sequence Read Archive PRJNA342805, 2017". In the Reference list, data citations must be labeled with "[DATASET]". A data reference must provide the database name, accession number/identifiers and a resolvable link to the landing page from which the data can be accessed at the end of the reference. Further instructions are available at <http://www.embopress.org/page/journal/14693178/authorguide#referencesformat>

10) Regarding data quantification (see Figure Legends:

<https://www.embopress.org/page/journal/14693178/authorguide#figureformat>)

- the name of the statistical test used to generate error bars and P values,

- the number (n) of independent experiments (please specify technical or biological replicates) underlying each data point,

- the nature of the bars and error bars (s.d., s.e.m.),

- If the data are obtained from a Program fragment delivered error ``Can't locate object method "less" via package "than" (perhaps you forgot to load "than"?) at //ejpvfs23/sites23b/embor_www/letters/embor_decision_revise_and_review.txt line 56.' 2, use scatter plots showing the individual data points.

12) Please also note our reference format:

13) All Materials and Methods need to be described in the main text using our 'Structured Methods' format, which is required for all research articles. According to this format, the Methods section includes a Reagents and Tools Table (listing key reagents, experimental models, software and relevant equipment and including their sources and relevant identifiers) followed by a Methods and Protocols section describing the methods using a step-by-step protocol format. The aim is to facilitate adoption of the methodologies across labs. More information on how to adhere to this format as well as a downloadable template (.docx) for the Reagents and Tools Table can be found in our author guidelines:

I look forward to seeing a revised version of your manuscript when it is ready. Please let me know if you have questions or comments regarding the revision.

Kind regards,

Deniz

Deniz Senyilmaz Tiebe, PhD
Senior Scientific Editor
EMBO Reports

Point-by-point Response:**Referee #1:**

This paper by Lyons et al. demonstrates that acyl-GIP improves lipid tolerance in obese male mice in a manner dependent on GIPR in *Ucp1*+ adipocytes.

This is a significant contribution to advancing our understanding of a medically relevant topic. The finding that brown adipose tissue (BAT) plays a metabolically protective role through pharmacological GIP is novel and underscores the need for further exploration of the therapeutic potential of brown fat. I have a few comments and suggestions that could strengthen this already compelling work.

We thank the reviewer for their positive feedback and enthusiasm for our study. However, we would like to clarify that we did not use a model that removes the GIPR in *UCP1*+ adipocytes exclusively, but rather one that reduces GIPR in *Myf5*+ cells. We have previously tried to generate a GIPR tissue specific KO model using the *UCP1*-Cre mice, but we were unsuccessful. This data was never published but it did contribute to our final paper that was published where we provide evidence using a reporter *Gipr*-Cre mouse model and single nucleus RNA sequencing to demonstrate where the GIPR is expressed in adipose tissue (in non-adipocyte fractions, not in adipocyte fractions; PMID: 35192688).

1. Line 122-124: Could the ~8% increase in fatty acid (FA) uptake by BAT be converted to absolute amounts, accounting for BAT volume/abundance? This could provide a clearer understanding of FA uptake and clearance at the whole-body level. Additionally, it may be useful to compare the delta between ³H-palmitic acid tracing into BAT in wild-type (WT) and BAT-specific GIPR knockout (KO) mice.

Thank you for this suggestion. By convention, these values are typically shown as a % of ingested activity, however we do agree that providing absolute values for ³H-palmitic acid uptake into BAT between GIP and PBS treated mice is important to demonstrate the importance of BAT GIPR signalling for lipid clearance at the whole-body level. Absolute values can be found in the supplemental figures.

We also appreciate your suggestion to repeat the ³H-palmitic acid uptake study with our BAT specific GIPR knockout model. We are in complete agreement that these results could provide direct evidence supporting our conclusions that GIP mediates lipid uptake in BAT through BAT GIPR agonism. We have an ongoing collaboration/set of experiments to tease apart GIPR signalling on BAT blood flow and lipid uptake, though we feel including those findings in this manuscript is outside of the scope of this study.

2. The hypothesis that GIP alters blood flow is intriguing. This could be easily tested using FITC-dextran, with or without acyl-GIP, in WT mice, which would help rule in or out this potential mechanism.

Thank you for providing a straightforward method to assess how GIP alters blood flow in BAT we appreciate your insight and knowledge of these techniques. We are currently working towards a more comprehensive understanding of how GIP can change blood flow in BAT through an active collaboration that will allow us to assess changes in blood flow and differences in metabolites taken up and released into circulation from BAT in response to GIP. This work will be part of a separate project and therefore, manuscript. As such, we believe that in addition to discussing the possibility of changes in blood flow as a potential mechanism driving changes in lipid uptake in lines 169-192, we will include the absence of blood flow data as a limitation to our study. We have added “As GIP has been previously demonstrated to alter adipose tissue blood flow, it may be possible that increases in GIP-mediated BAT uptake may have been a result of increased blood flow to the tissue.” in lines 234-236.

Referee #2:

This paper shows that acute treatment with GIP increases LPL activity and lipid uptake in BAT of obese mice. The effect of GIP on lipid tolerance was absent in whole body or BAT-selective GIPR knockout animals. RER values were also decreased by GIP treatment without a detectable change in energy expenditure or food intake. Interestingly, the effect of GIP on lipid clearance was not observed in chow fed (lean) animals. The experiments are topical given the great interest in understanding effects of GIP on body weight and metabolism. The studies are also technically sound. However, the paper is quite descriptive and could be strengthened by providing some mechanistic information regarding how GIP signaling may modulate BAT LPL activity, and why this may be selective to high fat diet/obesity conditions.

We would like to thank the reviewer for their insightful comments and suggestions, and appreciate their comment of how our studies are technically sound.

Additional comments follow:

1. Related to the mechanism by which GIP treatment increases LPL activity in BAT- have the authors monitored the LKB1-AMPK pathway in the BAT (vs. white fat depots) in their treatment paradigms?

Thank you for highlighting the importance of exploring the mechanism by which GIP treatment increases LPL activity in BAT and how it compares to WAT. The mechanisms by which GIP specifically influences LPL activity is quite complex and not much work has been completed in BAT. We do find this question falls outside of this current study, however, we are currently exploring this mechanism in both *in vitro* and *in vivo* models and plan to provide these data in a follow-up manuscript to this work.

2. Is there a change in GIPR expression in brown adipocytes or other BAT cell types from chow/lean vs. hfd/obese animals?

The reviewer brings up a great point and an important piece of data that would improve our study. To address this question, we have included data comparing mRNA expression of GIPR in

BAT, iWAT, eWAT, and rWAT from regular chow fed (lean) and HFD fed (obese) mice in **Fig. EV1K**. We now also discuss these results in lines **111-113**, where we say “There were no differences in adipose tissue *Gipr* mRNA expression between RCD and HFD fed mice ($p < 0.05$; BAT, eWAT, rWAT), aside from iWAT where GIPR expression significantly increased with HFD feeding ($p = 0.0165$; Fig. EV1K)”.

3. The effect on "whole body FAO" was driven by a decrease in the RER - this could reflect a switch towards fatty acid utilization in BAT due to increased lipid uptake. Did the authors assess if glucose uptake is lowered in the BAT following treatment?

We thank the reviewer for addressing this point regarding whole body FAO. Given the lack of changes in energy expenditure and based on how whole-body FAO is calculated, an increase in whole-body FAO would also mean a decrease in whole-body carbohydrate oxidation. A decrease in glucose uptake/increase in lipid uptake into BAT could possibly explain BAT GIPR mediated contributions to lipid tolerance if these substrates were being directly shuttled for oxidation. However, we don't believe that this is what is happening in response to GIP. In Fig. S2, we performed an LTT with the mice in metabolic chambers and we observed no changes in the rates of lipid oxidation. From this, we concluded that under the conditions of an LTT, GIP promotes lipid uptake, whereas under routine housing conditions GIP promotes lipid oxidation (Figure 3). We made an effort to separate our LTT and routine metabolic rate lipid oxidation data, however we may have not been clear about the distinction between how these two experiments were conducted and interpreted. To improve clarity, we moved Fig S2 into the main text as Fig. 3 and discuss these results in lines **144-148** expanding on the distinction between lipid uptake and oxidation in response to GIP as two important but different physiological responses to GIP. We have also changed the labelling in figures to state LTT (lipid tolerance test) and RMR (routine metabolic rate), so readers know the experimental conditions of the graphed results. We have also improved the clarity of our summary of findings in Figure 5, so that it separates out the mechanisms of lipid tolerance, uptake and whole-body fatty acid oxidation.

4. It would be interesting to know if the lipid clearance effects of GIP are diminished under thermoneutral conditions.

Thank you for this suggestion, we are currently preparing to conduct these experiments, however, the logistics of performing an LTT on thermoneutral mice is difficult. Since we perform experiments in a facility where the ambient temperature is 23°C, and we house mice at thermoneutrality (30°C), we have evidence to suggest that when the mice are moved from thermoneutrality to 23°C, they undergo a cold stress and mask any effects of GIP. Therefore, while we are completely interested in understanding the lipid clearance effects of GIP, the protocols/logistics to elucidate this mechanism is still being developed.

5. The Myf5Cre used for BAT KO would also eliminate GIPR in muscle and some other tissues during development. This should be discussed.

Thank you for bringing up this point. This would be a major oversight and invalidate our brown adipose tissue specific knockout model if the GIPR was expressed in skeletal muscle. However,

we have previously published (PMID: 31451430) that the GIPR is in fact not expressed in skeletal muscle nor in liver tissue. We do not detect any amounts of mRNA expression of the GIPR in our samples, and therefore, we do not believe that the Myf5-Cre: floxed GIPR model has any direct effects on GIPR signaling in skeletal muscle. To address any concerns or confusion we have added clarification about the model and the lack of skeletal muscle GIPR expression in lines 289-291, where we add “Despite *Myf5* being expressed in both skeletal muscle and BAT, skeletal muscle itself does not express GIPR, allowing this model to exclusively knockout GIPR in BAT.”

Referee #3:

This manuscript "Acute exogenous acyl-GIP administration enhances whole-body lipid handling and fatty acid oxidation via brown adipose tissue in obese male mice" reports a positive effect of an acylated glucose-dependent insulinotropic polypeptide (acyl-GIP) on lipid handling and fatty acid oxidation in mice. They further use GIPR inhibitor and Knockout mice to demonstrate that the effect of acyl-GIP depends on GIPR specifically in the brown adipocytes. The results have strong translational implication as it shows that GIPR signaling in brown fat underly GIP-induced systemic improvements of lipid metabolism. However, the data presented do not sufficiently support this major conclusion.

We thank the reviewer for their insightful comments/concerns and appreciate their identification that our findings have strong translational implications.

The first and foremost concern the Myf5Cre that was used to drive GIPR knockout in the brown adipocytes. The Cre driver is known to drive gene deletion in skeletal muscle progenitors (and therefore all muscle cells, PMID: 17540178), as well as other cell types in addition to muscle and fat (PMID: 20848654). As the skeletal muscle accounts for 40% of body weight and plays a key role in glucose and fatty acid metabolism, the role of muscle-specific GIPR signaling should have been analyzed to support the conclusion.

Thank you for this point, please see our response to query 5 from reviewer #2 as they had similar concerns.

The study overly relies on LTT data and a couple of related physiological assays, which are used to reflect the systemic response to lipid overload, but in depth molecular mechanistic analysis and function validation is lacking.

Thank you for this feedback. We are currently working on determining the molecular mechanisms driving the GIP mediated improvements in lipid tolerance and increases in lipid oxidation. Our goal of this manuscript was to present the previously undetermined but striking effects of long-acting GIP on whole body lipid metabolism through the BAT. Therefore, as previously mentioned in response to a reviewer's comment, we believe identifying these mechanisms, which could be quite extensive are beyond the scope of the current manuscript, we are currently working on these pathways within the BAT, and we will present this work in a future manuscript.

Minor Comments:

Methods often fall short of providing necessary information to others to repeat the work. Some sections have only 2 sentences. For instance: What is fatty acid in the acyl-GIP, how it was formulated, and what is purpose of this modification?

Thank you for highlighting the absence of some information from our methods, we agree that more clear description will improve our study. We have added additional information all throughout our methods, including addressing the instance you mention in your comment in lines 298-300 where we have included “In brief, palmitic acid was added to the N-terminus of a synthetic GIP analog to delay cleavage via dipeptidyl peptidase-4 inhibitor, extending the half life from 4-7 minutes (native GIP) to 6-10 hours (acyl-GIP).”

Dear Jacqueline,

Thank you for submitting your revised manuscript. It has now been seen by two of the original referees. My apologies for the delay in getting back to you, which was due to the delay in receiving referee reports.

As you will see, referees find that the study is significantly improved during revision and recommend publication. However, both referees have remaining minor outstanding concerns that need to be addressed. Referee #1 finds that it is necessary to understand which roles of GIP are adipose cell autonomous (point 1). Both referees find that the claims on the BAT specificity of the driver needs additional support (referee #2, first concern) and/or toning down (referee #1, point 2). As per the second comment of referee #2, it is not prerequisite to address for publication in EMBO Reports. Please also provide a point-by-point response. Please let me know if you would like to discuss any of the points further.

Moreover, the editorial points below need to be addressed before I can accept the manuscript.

- We find that your manuscript is better suited for Report format.
- Please provide 3-5 keywords for your study. These will be visible in the html version of the paper and on PubMed and will help increase the discoverability of your work.
- Please rename the Conflicts of Interest section as Disclosure and Competing Interests Statement, and move the section after Acknowledgements.
- Please specify author contributions by using the CRediT contributor role taxonomy in the journal submission system. Please see <https://www.embopress.org/page/journal/14693178/authorguide#authorshipguidelines> for further information.
- As per our format requirements, in the Reference section, DOIs should only be used for preprints and datasets that have not been published yet. Please remove the DOIs.
- Please fill out and include an author checklist as listed in our online guidelines (<https://www.embopress.org/page/journal/14693178/authorguide>)
- Funding information needs to be congruent in the manuscript and in the journal submission system. We note that the following is missing in the journal submission system as separate funders: Novo Nordisk-New Investigator Award, and Drucker Family Innovation Fund, Gales Family Charitable Foundation Postdoctoral Fellowship, Canadian Foundation for Innovation (CFI)-Ontario Research Fund (ORF) and CFI- John R. Evans Leaders Fund (JELF), NIH/NIDDK (DK046492, DK125353, DK123075, and DK141090).
- We note the following regarding the figure nomenclature/callouts: Fig. S2A-C is a wrong callout, it needs to be corrected; "Supplemental" should not be used as a callout. Please see <https://www.embopress.org/page/journal/14693178/authorguide#figureformat> for further information.
- All research articles submitted as revised versions must include a structured methods section that includes a Reagents and Tools Table followed by a Methods and Protocols section. Please see <https://www.embopress.org/page/journal/14693178/authorguide#structuredmethods> for further information.
- The count from the title page needs to be removed (word count, figure count, etc.).
- Our production/data editors have asked you to clarify several points in the figure legends - Figure Legends (main + EV):
 - o Please define the annotated p values ****/**/**/* as well as provide the exact p-values for the same in the legend of figure EV3 A as appropriate.
 - o Please note that the exact p values are not provided in the legends of figures 1B-E; 2C, G, H; 4B, F, H; EV1 A-H, K.
 - o Please note that information related to n is missing in the legend of figure 2H.
- Papers published in EMBO Reports include a 'synopsis' and 'bullet points' to further enhance discoverability. Both are displayed on the html version of the paper and are freely accessible to all readers. The synopsis includes a short standfirst summarizing the study in 1 or 2 sentences (max 35 words) that summarize the paper and are provided by the authors and streamlined by the handling editor. I would therefore ask you to include your synopsis blurb and 3-5 bullet points listing the key experimental findings.
- In addition, please provide an image for the synopsis. This image should provide a rapid overview of the question addressed in the study but still needs to be kept fairly modest since the image size cannot exceed 550 (width) x 300-600 (height) pixels.

Thank you again for giving us to consider your manuscript for EMBO Reports, I look forward to your minor revision.

Kind regards,

Deniz

--

Deniz Senyilmaz Tiebe, PhD
Senior Scientific Editor
EMBO Reports

Referee #1:

The authors have addressed many of the reviewers' comments. There remain limitations in the mechanistic depth provided in the paper. However, the paper is technically sound and provides important new information for the field. I have two points for authors to consider prior to publication.

1-Given the discussed and previously reported role of GIP in regulating blood flow (and related to concern about specificity of the Cre driver), it would be relevant to understand which, if any, of the effects observed are adipose cell autonomous. Does GIP treatment regulate lipid uptake, LPL etc in isolated brown preadipose or adipocytes?

2-As multiple reviewers noted, the Myf5-Cre is not specific to BAT: it is expressed in multiple tissues, including some white fat depots and certain brain regions. The authors should not refer to this model as "BAT-specific".

Referee #2:

I only raised two main comments in the previous round but neither of these was addressed. The first concern was the Myf5-Cre used to drive Gipr KO could also target the skeletal muscle and other cell types. The authors responded that Gipr is not expressed in the skeletal muscle, citing a paper published earlier in Mol Metab by the same group. However, there was no data provided so I tried to find the information in that paper, without luck. No discussion or rationale of choosing Myf5-Cre was in the manuscript (other than a brief statement in the methods without citing any papers). Some validation of Gipr expression in the Myf5-lineage cells in WT and KO mice should be provided. My second comment was about the lack of mechanisms that other reviewers also mentioned. The authors responded that it was beyond the scope of the work, which I don't agree with - given this is an Embo brand journal that typically have high bars. The authors could have simply validated some known targets of the pathway and added some discussion to the caveat/pitfall paragraph.

Nutritional Sciences
UNIVERSITY OF TORONTO

August 8th, 2025

RE: Minor revisions to *EMBO Reports*

Dear Dr. Deniz Senyilmaz Tiebe,

We have carefully reviewed and responded to the reviewers' comments and believe that we have adequately addressed all the concerns raised. Notably, in the revised version referee # 1 praised the article to say that it was “*technically sound provides new information for the field*”. They also raised some points about the observation that GIP increased LPL activity in brown adipose tissue and was curious if this effect was an adipose cell autonomous mechanism with GIP administration or if it could be related to the changes in blood flow. Therefore, in our response to the referees that we have addressed these comments directly by providing some more preliminary data to the referee. **[REDACTED: reference to unpublished data]**.

In addition, we also agree with referee #1 in that we can be more specific in our description of the *Myf5*-Cre mouse model and describe it as such instead of labeling it as a BAT-specific GIPR mouse model. We have updated our manuscript to reflect these changes.

Lastly, in response to Referee #2 comments about the validation of the *Myf5*-lineage cells in wildtype and knockout *Gipr* mice tissues we have gone ahead and processed the *Gipr* mRNA expression in skeletal muscle from our control and knockout mice and added this in an updated data set in Figure EV3A. As similarly published by our group in 2019 in *Molecular Metabolism* in Figure 2A, we found that *Gipr* mRNA expression to be not detectable in the liver or skeletal muscle.

We truly appreciate the reviewer's comments by this second round of revisions, and we hope that our responses have satisfied the points raised.

Sincerely,

Jacqueline Beaudry, PhD

Department of Nutritional Sciences, Temerty Faculty of Medicine, University of Toronto
Medical Sciences Building, 1 King's College Circle, 5th Floor, Room 5342, Toronto, Ontario M5S 1A8 Canada
P: 416-978-6527 | E: jacqueline.beaudry@utoronto.ca | W: <https://nutrisci.med.utoronto.ca>

Dr. Jacqueline Beaudry
University of Toronto
Nutritional Sciences
1 KINGS COLLEGE CIRCLE
TORONTO, ON M5S 1A8
Canada

Dear Jacqueline,

Thank you for submitting your revised manuscript and swiftly addressing the additional points later. I have now looked at everything and all is fine. Therefore, I am very pleased to accept your manuscript for publication in EMBO Reports.

Congratulations on a nice work!

Kind regards,

Deniz

--

Deniz Senyilmaz Tiebe, PhD
Senior Scientific Editor
EMBO Reports

--
